# Models Under SCOPE: Scalable and Controllable Routing via Pre-hoc Reasoning

Qi Cao [* 1]  Shuhao Zhang [* 1]  Ruizhe Zhou [2]  Ruiyi Zhang [1]  Peijia Qin [1]  Pengtao Xie [1]

## Abstract

Model routing chooses which language model to use for each query. By sending easy queries to cheaper models and hard queries to stronger ones, it can significantly reduce inference cost while maintaining high accuracy. However, most existing routers treat this as a fixed choice among a small set of models, which makes them hard to adapt to new models or changing budget constraints. In this paper, we propose SCOPE (**S**calable and **C**ontrollable **O**utcome **P**erformance **E**stimator), a routing framework that goes beyond model selection by predicting their cost and performance. Trained with reinforcement learning, SCOPE makes reasoning-based predictions by retrieving how models behave on similar problems, rather than relying on fixed model names, enabling it to work with new, unseen models. Moreover, by explicitly predicting how accurate and how expensive a model will be, it turns routing into a dynamic decision problem, allowing users to easily control the trade-off between accuracy and cost. Experiments show that SCOPE is more than just a cost-saving tool. It flexibly adapts to user needs: it can boost accuracy by up to **25.7%** when performance is the priority, or cut costs by up to **95.1%** when efficiency matters most. We release the dataset and code at our project page: `https://sullivan07043.github.io/SCOPE/`.

## 1. Introduction

Reasoning Language Models (RLMs) (OpenAI, 2025b; DeepSeek, 2025) have achieved remarkable performance, but they come with high costs and latency (Wei et al., 2023; Shinn et al., 2023; Wang et al., 2023b; Cao et al., 2025).

---
[*]Equal contribution  [1]University of California, San Diego [2]Independent Researcher. Correspondence to: Pengtao Xie <p1xie@ucsd.edu>.

*Proceedings of the 43rd International Conference on Machine Learning*, Seoul, South Korea. PMLR 306, 2026. Copyright 2026 by the author(s).

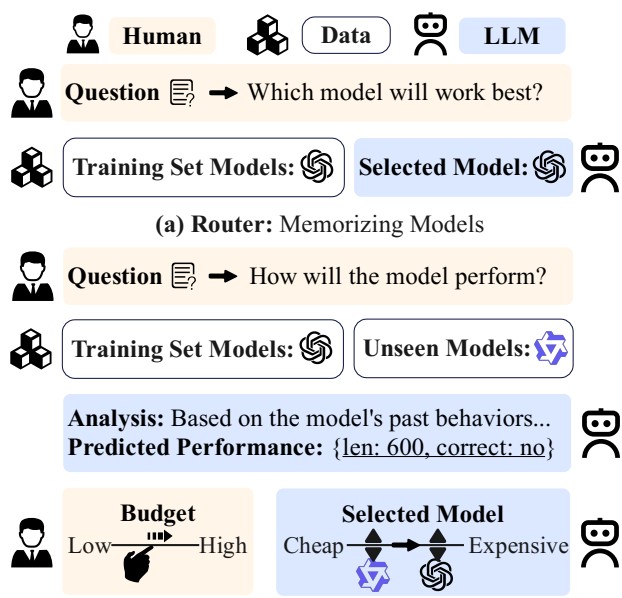

(a) **Router:** Memorizing Models

(b) **SCOPE:** Reasoning on Model Behaviors

*Figure 1.* **Paradigm Comparison.** Unlike other LLM-based routers (a) that perform closed-set classification, SCOPE (b) leverages the model's past behaviors to explicitly **predict token length and correctness**. This pre-hoc estimation enables **generalization to unseen models** and facilitates **budget-aware decision-making.**

In real-world applications, query difficulty varies significantly—using a powerful, expensive model for every simple query is wasteful. To solve this, model routing (Chen et al., 2023; Ong et al., 2025) adopts a portfolio approach: it assigns simple queries to cheaper models and complex problems to more capable ones. The goal is to find the best balance between performance and cost for each input.

However, current routers face two limitations because they treat routing as a simple classification task (i.e., selecting a model from a fixed list). First, they cannot handle new models. Traditional routers learn to choose from a specific set of models. If a new model is released, the router does not know how to use it. Integrating a new model requires collecting new data and retraining the entire system, which is inefficient in the rapidly evolving LLM landscape. Second, they lack flexible control. Most routers make a "hard" decision (e.g., simply outputting "Model A") without explaining why. They do not estimate how much better or

more expensive a model is. As a result, users cannot adjust the router's behaviors dynamically, for example, to save money when the budget is tight or to maximize accuracy for high-stakes tasks.

To overcome these limitations, we propose SCOPE (**S**calable and **C**ontrollable **O**utcome **P**erformance **E**stimator), which changes the routers in two fundamental ways, as shown in Fig. 1. First, we move from fixed candidates to behavioral fingerprints. Traditional routers just memorize model names, but SCOPE makes decisions by looking at how a model actually answers similar questions—its "fingerprint". By retrieving these behavioral signatures, SCOPE can evaluate any model, even one it has never seen before, simply by reading its fingerprint. This enables generalization to new models without retraining. Second, we shift from simple selection to detailed estimation. Instead of just saying "Pick Model A", SCOPE predicts if a model will be correct and estimates how many tokens it will cost. After that, the final decision is made by maximizing a utility function that weighs accuracy against cost, adjusted dynamically by the user's budget preference. This transforms routing into a flexible optimization task, allowing users to seamlessly navigate the trade-off between performance and cost.

Our contributions are summarized as follows:

- We show that model routing does not have to be a simple choice among a fixed set of models. Instead, SCOPE predicts how well a model will perform and how much it will cost based on its past behaviors on similar questions, which allows the system to naturally handle new models without retraining.

- SCOPE gives users direct control over the trade-off between accuracy and cost. By using predicted performance and cost, users can decide whether to save money or achieve higher accuracy for each query, without changing the system itself.

- We demonstrate that SCOPE consistently achieves better accuracy–cost trade-offs than any individual model in the evaluated pool. Depending on user preferences, it can either improve accuracy by up to **25.7%** or reduce inference cost by up to **95.1%**, making it an effective way to allocate test-time compute.

## 2. Related Work and Preliminaries

**Predictive Routing Methods.** A line of work approaches model routing through lightweight predictive signals. Methods such as Avengers-Pro (Zhang et al., 2025b) and OmniRouter (Mei et al., 2025) rely on embedding-based retrieval or clustering to guide fast model selection, while Route-To-Reason (RTR) (Pan et al., 2025) and CARGO (Barrak et al., 2025) further incorporate learned regressors to estimate performance- or cost-related scores. These approaches expose explicit continuous signals (e.g., confidence or predicted utility), enabling controllable routing via optimization and thresholding. However, such systems often require carefully tuned feature extractors, regressors, and decision thresholds, resulting in a multi-stage pipeline with substantial hyperparameter sensitivity. This complexity makes deployment and adaptation difficult.

**LLM-based Routers.** Another line of work leverages Large Language Models that directly map input queries to routing decisions. Methods such as Router-R1 (Zhang et al., 2025a) and xRouter (Qian et al., 2025) train LLMs with reinforcement learning (e.g., PPO (Schulman et al., 2017) or DAPO (Yu et al., 2025)) to internalize routing policies, while CoRL (Jin et al., 2025) and Reasoning Router 0.6B (Mohseni, 2025) rely on prompt-based reasoning or difficulty classification. LLM-based routers benefit from strong natural language understanding, rich internal knowledge, and robust instruction-following ability. These properties allow them to interpret complex, loosely specified constraints, leverage prior knowledge about task difficulty or model behaviors, and provide a precise decision.

**Motivation: Harmonizing Flexibility and Reasoning.** SCOPE is designed to combine the strengths of two different routing philosophies. On one hand, it retains the ability of large language models to understand complex questions and generate step-by-step reasoning. On the other hand, it allows users to clearly control how much accuracy or cost they are willing to trade for a given query. Instead of committing to a fixed decision rule, SCOPE first assesses how well a model is likely to perform and how expensive it will be, and only then decides which model to use. This separation between prediction and decision-making enables SCOPE to be both powerful in reasoning and flexible in practice.

**Roadmap.** We now formalize the above design principles and describe how they are implemented in SCOPE, as illustrated in Fig. 2. Section 3 addresses generalization by introducing retrieval-augmented (Lewis et al., 2021) model fingerprinting (Fig. 2a), enabling training-free adaptation to unseen models. Section 4 focuses on prediction reasoning model training, detailing a post-training pipeline based on hindsight distillation (Liu et al., 2023) and Group Relative Policy Optimization (GRPO) (Shao et al., 2024) (Fig. 2b). Finally, Section 5 establishes controllability through an anchor-calibrated utility function that translates pre-hoc predictions into inference-time decisions (Fig. 2c).

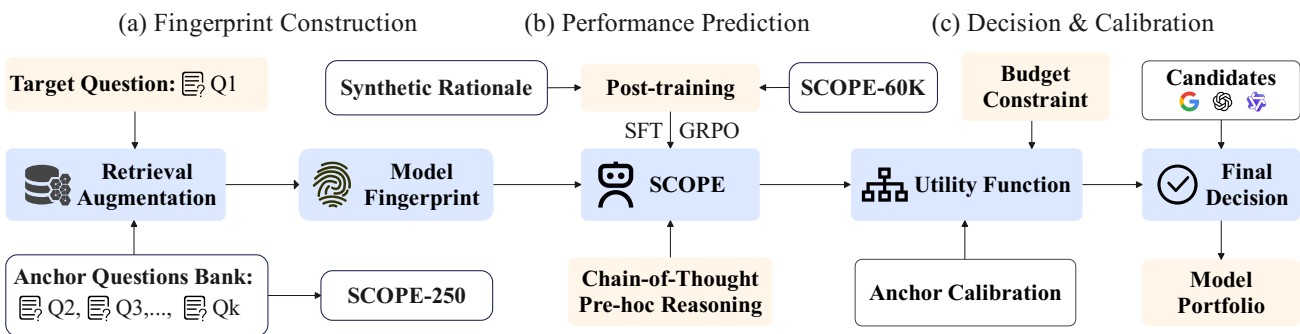

*Figure 2.* **The SCOPE Framework.** The pipeline consists of three stages: (a) constructing behavioral fingerprints via anchor retrieval; (b) estimating outcome correctness and cost using a reasoning-driven predictor optimized with SFT and GRPO; and (c) selecting the optimal model via a calibrated, budget-aware utility function. Ablation studies of key components are detailed in Section. 6.2.

## 3. Fingerprint Construction

SCOPE casts pre-hoc performance estimation as a retrieval-augmented, in-context simulation problem. Given a target query $x$ and a target model $M$, rather than inspecting model weights, we retrieve task-relevant behavioral patterns from a pre-computed "fingerprint" to predict the model's correctness and cost.

### 3.1. Model Fingerprinting

To characterize models universally without retraining, we employ a fixed **Anchor Set** $\mathcal{B} = \{x_i\}_{i=1}^N$, consisting of $N$ representative queries covering diverse task distributions.

For any specific model $M$, we generate a **Model Fingerprint** $\phi_{\mathcal{B}}(M)$ by recording its ground-truth performance on the anchor set:

$$\phi_{\mathcal{B}}(M) = \left\{ (x_i, y_i^M, c_i^M) \right\}_{i=1}^N \quad (1)$$

where $y_i^M \in \{0, 1\}$ denotes correctness and $c_i^M \in \mathbb{R}_+$ denotes token cost. This mechanism enables training-free scalability: adapting SCOPE to a new model requires only a single forward pass over $\mathcal{B}$ to generate its fingerprint, enabling estimation on arbitrary future queries without gradient updates.

### 3.2. Retrieval-Augmented Context

For a target query $x_{\text{target}}$, we identify the most relevant historical evidence using dense retrieval. Let $s(x, q)$ be the cosine similarity between the embeddings of two queries[1]. We retrieve the top-$K$ anchors:

$$\mathcal{A}_K(x_{\text{target}}) = \text{TopK}_{x_i \in \mathcal{B}} \ s(x_{\text{target}}, x_i) \quad (2)$$

We then extract the corresponding performance metrics for model $M$ to form the retrieved fingerprint slice:

$$\phi_K(x_{\text{target}}, M) = \{(x_i, y_i^M, c_i^M) \mid x_i \in \mathcal{A}_K(x_{\text{target}})\} \quad (3)$$

---

[1]Qwen3-Embedding-0.6B (Zhang et al., 2025c) is used.

Finally, we serialize this slice into a structured text format to condition the estimator. The final prompt is constructed as:

$$P(x_{\text{target}}, M) = I \parallel \text{Ser}(\phi_K(x_{\text{target}}, M)) \parallel x_{\text{target}} \quad (4)$$

where $I$ is the system instruction and $\parallel$ denotes string concatenation. This allows SCOPE to infer the likely outcome for $x_{\text{target}}$ by analyzing how model $M$ behaved on semantically similar problems.

## 4. Performance Prediction and Model Training

We instantiate SCOPE as a reasoning language model designed to provide robust performance estimates grounded in interpretable rationales.

### 4.1. Reasoning Estimator

Conditioned on the retrieval-augmented prompt $P(x, M)$ containing the query $x$ and model $M$, the estimator generates a reasoning path $z$ followed by a structured prediction tuple. Formally, we sample a response sequence:

$$z, (\hat{y}, \hat{\ell}) \sim p_\theta(\cdot \mid P(x, M)) \quad (5)$$

where $z$ is the chain-of-thought rationale, $\hat{y} \in \{0, 1\}$ is the predicted binary correctness, and $\hat{\ell} \in \mathbb{R}_+$ estimates the token consumption. By grounding predictions in $z$, SCOPE ensures estimates are causally derived from observed behavioral patterns in the fingerprint rather than mere statistical guessing.

### 4.2. Data Construction

To align SCOPE with diverse reasoning behaviors, we construct two complementary datasets.

**SCOPE-60K (Supervision).** We collect a comprehensive dataset of model-query interactions across 13 diverse LLMs, covering domains from STEM to Humanities (Fig. 3). Each

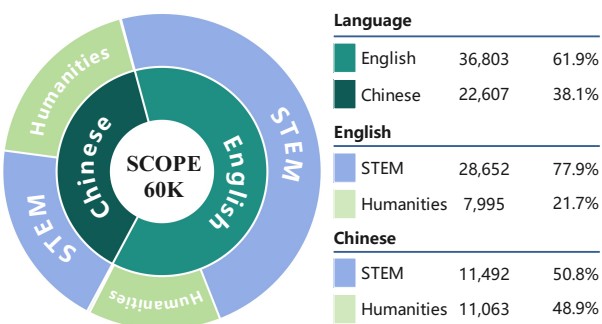

| Language | | |
|---|---|---|
| ■ English | 36,803 | 61.9% |
| ■ Chinese | 22,607 | 38.1% |
| **English** | | |
| ■ STEM | 28,652 | 77.9% |
| ■ Humanities | 7,995 | 21.7% |
| **Chinese** | | |
| ■ STEM | 11,492 | 50.8% |
| ■ Humanities | 11,063 | 48.9% |

*Figure 3.* **Composition of SCOPE-60K.** The dataset spans diverse domains, primarily STEM (e.g., Math 29.7%, Chemistry 20.2%, Physics 18.0%, etc) and Humanities (e.g., History 21.4%, Politics 20.7%, Chinese 18.0%, etc).

sample $(x, M, y, \ell)$ records the input query, model identity, ground-truth correctness, and token usage, serving as the basis for our reward modeling.

**SCOPE-250 (Fingerprinting).** We curate a compact anchor set of 250 representative queries. This set functions as a "topological skeleton" of the task space, selected to preserve the category distribution of SCOPE-60K(Fig. 12 in Appendix G ). This ensures that the retrieved fingerprints provide in-context evidence that is distributionally aligned with test-time queries.

### 4.3. Training Strategy

We perform a two-stage training of SCOPE, where each phase serves a distinct purpose. The **SFT stage** primarily initializes the model to produce concise CoT outputs; this ensures inference efficiency and yields cheaper, more stable rollouts for the subsequent reinforcement learning phase. Building on this efficient base, the **GRPO stage** serves as the main driver for maximizing predictive accuracy.

**Stage 1: SFT via Hindsight Distillation.** We first perform Supervised Fine-Tuning (SFT) using a *hindsight distillation* (Liu et al., 2023) approach. We prompt a teacher model with both the query and the ground-truth outcomes $(y, \ell)$ to generate a coherent, concise Chain-of-Thought $z$ (CoT) (Wei et al., 2023) that justifies the realized metrics. The model is then trained to predict $z, \hat{y}, \hat{\ell}$ using standard next-token prediction.

**Stage 2: Reinforcement Learning via GRPO.** To enforce output formatting and improve estimation precision, we further align the model using GRPO. We define a gated composite reward function:

$$R(o) = \mathcal{G}(o) \cdot \left( R_{\text{corr}}(\hat{y}, y_{\text{gt}}) + R_{\text{token}}(\hat{\ell}, \ell_{\text{gt}}) \right) \quad (6)$$

where $\mathcal{G}(o)$ is a binary format gate.

- **Correctness Reward** ($R_{\text{corr}}$): A sparse reward that equals 1 if the predicted binary label matches the ground truth, and 0 otherwise.

- **Adaptive Token Reward** ($R_{\text{token}}$): To handle variance in model verbosity, we design an **adaptive tolerance** mechanism. For reasoning models, we use a relative error tolerance (e.g., within 50%); for standard models, we use a tighter absolute tolerance. (See Appendix B.2 for the full formulation).

## 5. Decision and Calibration

Given a candidate set $\mathcal{M} = \{M_1, \dots, M_K\}$, our objective is to select the optimal model $M^*$ for each query $q$ that maximizes a user-defined trade-off between performance and cost. We formulate this as a utility maximization problem governed by a preference coefficient $\alpha \in [0, 1]$, where $\alpha = 1$ prioritizes accuracy and $\alpha = 0$ prioritizes cost.

### 5.1. Utility Formulation

To unify accuracy and cost into a single metric, we first apply min-max normalization within question clusters to ensure comparability. For cost normalization, we employ a log-transformation to enhance sensitivity in low-cost regimes. We define the predicted utility $U_{\text{pred}}(M_i)$ based on SCOPE's output:

$$U_{\text{pred}}(M_i) = \alpha \cdot \hat{y}_i + (1 - \alpha) \cdot (1 - \hat{c}_i)^{\gamma_{\text{dyn}}} \quad (7)$$

where $\hat{y}_i$ is the predicted binary correctness, $\hat{c}_i$ is the normalized predicted cost, and $\gamma_{\text{dyn}}$ is a sensitivity factor.

### 5.2. Anchor-Based Calibration

To improve robustness, we incorporate a calibration signal derived from the ground-truth performance of retrieved anchors. This acts as a historical prior. For the top-$k$ retrieved anchors, we compute an aggregated calibration utility $U_{\text{cal}}(M_i)$ using the weighted average of their actual performance and costs, weighted by their semantic similarity to the query.

### 5.3. Final Decision

The final routing decision aggregates the real-time prediction ($U_{\text{pred}}$) and the historical calibration ($U_{\text{cal}}$):

$$M^* = \arg \max_{M_i \in \mathcal{M}} \left( w \cdot U_{\text{cal}}(M_i) + (1 - w) \cdot U_{\text{pred}}(M_i) \right) \quad (8)$$

where $w \in [0, 1]$ is an adaptive weighting factor that balances the influence of historical priors versus instance-specific predictions. To ensure smooth transitions across the Pareto frontier, we design $w$ to adjust dynamically based on the strictness of the budget constraint $\alpha$. We provide the

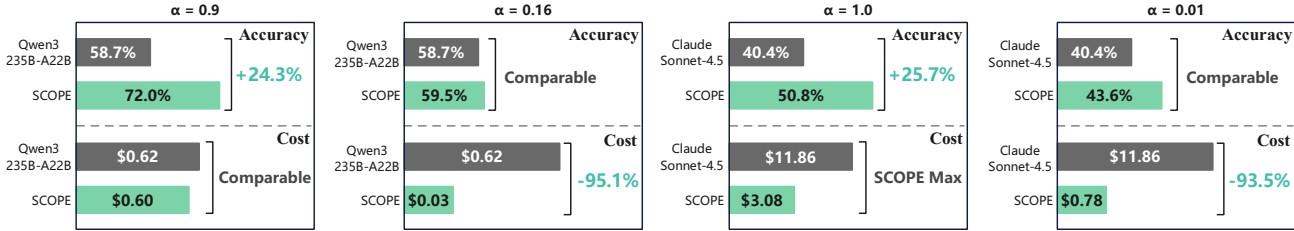

*Figure 4.* **Comparison with Individual Models.** SCOPE shows effective trade-off performance in two distinct regimes. **Performance Boosting:** With high $\alpha$, SCOPE transcends the accuracy ceiling of strong baselines (`Qwen3-235B` and `Claude-Sonnet-4.5`) by over **+24%** while maintaining comparable or lower costs. **Cost Efficiency:** With low $\alpha$, SCOPE maintains competitive accuracy while slashing inference costs by up to **95.1%**. More models' results are provided in Appendix Fig. 13.

*Table 1.* Routing performance comparison. We evaluate SCOPE under different trade-off coefficients $\alpha$ against baseline routers on both Test Set and OOD Set.

| Setting | Test Set | | | OOD Set | | |
|---|---|---|---|---|---|---|
| | PGR (%, ↑) | Avg. A (%, ↑) | Cost ($, ↓) | PGR (%, ↑) | Avg. A (%, ↑) | Cost ($, ↓) |
| Oracle | 100.00 | 83.33 | 0.09 | 100.00 | 64.00 | 0.98 |
| Random | 50.75 | 57.14 | 0.65 | 33.00 | 41.60 | 3.59 |
| Cheapest | 41.79 | 52.38 | 0.02 | 23.43 | 38.40 | 0.46 |
| Most Expensive | 65.67 | 65.08 | 2.50 | 31.80 | 41.20 | 11.89 |
| SVM Router (Feng et al., 2025b) | 0.00 | 30.16 | 0.01 | 29.41 | 40.40 | 11.86 |
| MLP Router (Hu et al., 2024) | 26.12 | 44.05 | 0.28 | 31.80 | 41.20 | 11.52 |
| kNN Router (Hu et al., 2024) | 10.45 | 35.71 | 0.16 | 29.41 | 40.40 | 11.79 |
| Graph Router (Feng et al., 2025a) | 80.60 | 73.02 | 0.72 | 34.20 | 42.00 | 1.37 |
| xRouter (Qian et al., 2025) | / | 57.94 | 1.01 | / | 30.57 | 1.16 |
| SCOPE ($\alpha = 0.0$) | 44.03 | 53.57 | 0.03 | 27.02 | 39.60 | 0.75 |
| SCOPE ($\alpha = 0.6$) | 67.91 | 66.27 | 0.22 | 53.34 | 48.40 | 1.38 |
| SCOPE ($\alpha = 1.0$) | **84.33** | **75.00** | 0.63 | **59.32** | **50.80** | 3.08 |

detailed formulation of this dynamic weighting mechanism in Appendix B.3.

# 6. Experiments

In this section, we evaluate SCOPE through a set of experiments. Our evaluation focuses on three main questions:

**Q1. Routing Performance:** Can SCOPE achieve a better accuracy–cost trade-off than every individual model and existing baselines?

**Q2. Ablation Study:** How does each key component contribute to the overall performance of the system?

**Q3. Cost Analysis:** Does SCOPE significantly reduce costs compared to other methods?

**Experimental Setup.** We evaluate our framework using a diverse portfolio of 11 models, categorized into 7 *seen* models (used for training) and 4 *unseen* models (held out for training-free generalization; see Appendix Tab. 8). Our evaluation is conducted across two distinct regimes: (1) **In-Distribution (Test Set):** To assess standard routing ef-

fectiveness, we utilize a held-out 5% split of the SCOPE-60K dataset. This set encompasses diverse tasks sourced from established benchmarks, including MMLU-Pro (Wang et al., 2024), R-Bench (Guo et al., 2025), GAOKAO (Zhang et al., 2024), and GPQA (Rein et al., 2023). (2) **Out-of-Distribution (OOD Set):** To stress-test robustness against unseen architectures and frontier-level difficulty, we curate a specialized set of 250 queries. These samples are drawn exclusively from the most challenging reasoning benchmarks: AIME (2024/2025) (Mathematical Association of America, 2024–2025), Humanity's Last Exam (Phan et al., 2025), SimpleQA (Wei et al., 2024), and OlympiadBench (He et al., 2024).

## 6.1. Routing Performance (Q1)

To answer Q1, we benchmark SCOPE against both individual models and existing router baselines. We perform these comparisons under two settings: the Test Set, which evaluates performance on seen models; and the OOD Set, which tests if the system can generalize to new, unseen models.

**Comparison with Individual Models.** One key result is that SCOPE can outperform individual models while re-

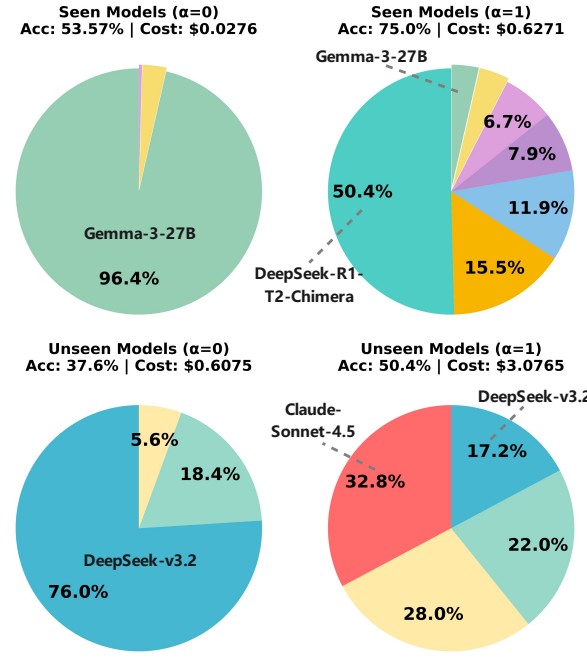

*Figure 5.* **Adaptive Model Portfolio.** SCOPE autonomously reconfigures its selection based on $\alpha$. It transitions from low-cost dominance at $\alpha = 0$ to a diversified portfolio at $\alpha = 1$. More details are provided in Appendix Fig. 14.

maining cost-efficient. As shown in Fig. 4, SCOPE behaves as a dynamic system that consistently achieves better accuracy–cost trade-offs than any single static model. When accuracy is prioritized ($\alpha \approx 1.0$), SCOPE improves accuracy by +24.3% over the strong training-set model `Qwen3-235B-A22B`. More importantly, on the unseen model `Claude-Sonnet-4.5`, it improves accuracy by +25.7% while reducing inference cost by 74% ($3.08 vs. $11.86).

We further analyze this effect using Pareto frontier analysis. As illustrated in Fig. 6 (left), individual models correspond to fixed operating points, while SCOPE forms a dynamic frontier that covers a wider range of accuracy–cost trade-offs. These results show that SCOPE is not only a way to save cost, but also a practical form of pre-hoc test-time scaling (TTS) (Snell et al., 2024): by predicting which queries are difficult and allocating stronger models accordingly, it allows the combined system to outperform any single model.

**Comparison with Router Baselines.** Tab. 1 reports the results compared with both predictive routing methods and LLM-based routers. We report Average Accuracy (Avg. A), total cost, and Performance Gap Recovered (PGR) (Ong et al., 2025), which measures how close a router gets to the optimal choice (i.e., the cheapest model that can answer the query correctly).

**On the Test Set**, all supervised baselines are trained under the same conditions: SVM (Hearst et al., 1998), KNN (Kearns, 1989), MLP (Popescu et al., 2009), and Graph Router (Feng et al., 2025a) use the same SCOPE-60K split as our method. Under this setting, SCOPE ($\alpha = 1.0$) achieves 75.00% accuracy, clearly outperforming the strongest supervised baseline. We also compare with xRouter (Qian et al., 2025), which uses a different model pool with proprietary high-end models (e.g., `GPT-5-mini`, `o4-mini`). Even with this advantage, xRouter reaches only 57.94% accuracy at a cost of $1.01. In contrast, SCOPE improves accuracy by over 17% while reducing cost by 38%.

**On the OOD Set**, the ability to handle unseen models becomes crucial. xRouter keeps a fixed candidate pool and therefore performs poorly on these benchmarks. For supervised baselines, we re-train them using the anchor set (SCOPE-250) so that they can include the new models as labels. Even with this adaptation, their performance drops to near random due to the distribution shift. In contrast, SCOPE generalizes to unseen models without retraining. By retrieving past behaviors of similar models, it can effectively route queries and maintain strong performance where traditional classifiers and fixed policies fail.

**Model Portfolio.** Fig. 5 shows how SCOPE adapts its routing behaviors under different settings. When efficiency is prioritized ($\alpha = 0$), the system mainly selects low-cost models (e.g., assigning 76.0% of queries to `DeepSeek-v3.2`). When accuracy is prioritized ($\alpha = 1$), it instead shifts toward stronger and more capable models. Importantly, on the OOD set, SCOPE can identify and use powerful new models (e.g., `Claude-Sonnet-4.5`) that were not seen during training, showing that our fingerprinting method can capture model quality from past behaviors without requiring retraining.

### 6.2. Ablation Study (Q2)

In this section, we break down SCOPE to understand how its main components contribute to overall performance. Specifically, we study three questions: (1) How accurate is SCOPE at predicting model performance? (2) Does chain-of-thought reasoning help improve these predictions? (3) How do the utility function and anchor-based calibration affect the final routing results?

**Accuracy of Pre-hoc Estimation.** We implement SCOPE by fine-tuning the `Qwen3-4B-Instruct-2507` (Qwen, 2025) backbone (referred to as `Qwen3-4B` in this paper). To evaluate how accurate our pre-hoc predictions are, we compare SCOPE with standard baselines and with a non-reasoning variant on the test set. To isolate the effect of reasoning, we train a variant called SCOPE$_{NoCoT}$, using a dataset that removes reasoning traces, so that the model

*Table 2.* Predictive accuracy across different categories. We report Mean Absolute Error (MAE) for token length and Accuracy (ACC) for correctness. SCOPE variants consistently outperform the base model without training.

| Method | Anchors | Mathematics | | Physics | | Chemistry | | History | | Engineering | | Overall | |
|---|---|---|---|---|---|---|---|---|---|---|---|---|---|
| | | MAE ↓ | ACC ↑ | MAE ↓ | ACC ↑ | MAE ↓ | ACC ↑ | MAE ↓ | ACC ↑ | MAE ↓ | ACC ↑ | MAE ↓ | ACC ↑ |
| SCOPE | 5 | **2085** | 69.9% | **2307** | **79.9%** | **2452** | 78.4% | 413 | 67.1% | **3112** | **78.7%** | **1555** | **77.0%** |
| SCOPE$_{NoCoT}$ | 5 | 2339 | **72.5%** | 2643 | 74.3% | 2991 | **83.7%** | 393 | **70.5%** | 3113 | 71.0% | 1739 | 75.1% |
| Qwen3-4B | 5 | 2292 | 62.5% | 2630 | 55.6% | 2677 | 66.3% | 631 | 60.6% | 3391 | 59.4% | 1740 | 59.4% |
| Qwen3-4B$_{NoCoT}$ | 5 | 2339 | 58.3% | 2649 | 45.0% | 2613 | 54.7% | 508 | 62.2% | 3513 | 59.8% | 1728 | 56.7% |
| Qwen3-4B | 0 | 2997 | 61.3% | 3559 | 37.9% | 3294 | 38.8% | 674 | 61.5% | 4477 | 53.1% | 2213 | 53.5% |
| Qwen3-4B$_{NoCoT}$ | 0 | 3046 | 61.4% | 3587 | 40.5% | 3321 | 42.6% | 705 | 61.5% | 4526 | 56.6% | 2234 | 54.3% |

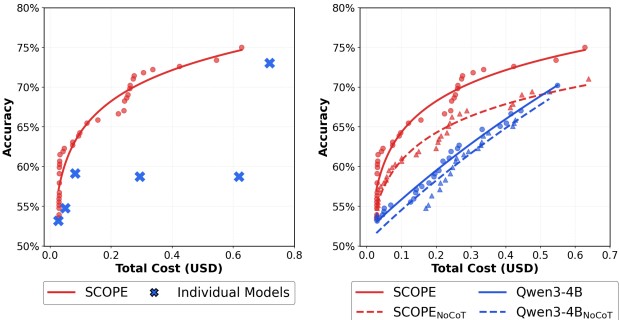

*Figure 6.* **Effectiveness of SCOPE and Reasoning Strategy.** (Left) SCOPE establishes a superior Pareto frontier compared to individual models, consistently achieving higher accuracy at lower costs. (Right) Ablation results demonstrate that our RL-enhanced CoT strategy (Solid Red) significantly outperforms direct prediction baselines (Dashed lines), validating that reasoning capability is essential for precise performance estimation.

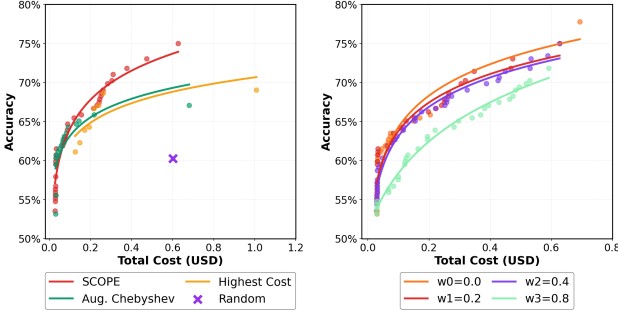

*Figure 7.* **Efficacy of Utility Function Design.** (Left) Our dynamic utility maximization strategy significantly outperforms standard scalarization baselines (e.g., Chebyshev, Random), proving the effectiveness of the proposed routing objective. (Right) Sensitivity analysis on the aggregation weight $w$ confirms the rationality of our two-stage design: balancing historical anchor statistics with real-time predictions yields the most robust trade-off curve.

learns a direct mapping from input to output. As shown in Tab. 2, SCOPE achieves the highest accuracy of 77.0% and the lowest token prediction error (MAE 1555). This clearly outperforms the few-shot baseline (Qwen-4B at 59.4%), showing the effectiveness of our training approach. Moreover, SCOPE also outperforms SCOPE$_{NoCoT}$ (75.1%), indicating that generating intermediate reasoning steps helps ground the prediction and improves estimation accuracy.

**Impact of Reasoning and Training Strategy.** To understand the impact of reasoning and training strategy on routing performance, we compare different methods on the cost–accuracy Pareto frontier, as shown in Fig. 6 (right). First, both SCOPE variants (red lines) clearly outperform the untuned Qwen3-4B baseline (blue lines) across all cost levels. This shows that the base model cannot reliably predict performance through simple prompting alone, and that our training pipeline is necessary to learn this ability. Second, among the trained variants, the reasoning-based SCOPE (solid red line) consistently outperforms SCOPE$_{NoCoT}$ (dashed red line). This suggests that while fine-tuning provides the basic capability, adding reasoning supervision helps the model make more precise predictions,

leading to better routing decisions.

**Effectiveness of Calibration and Decision Logic.** Finally, we examine our decision-making framework by studying the utility function and the calibration prior in Fig. 7. For the utility function (Left), our dynamic maximization strategy consistently outperforms standard baselines across the Pareto frontier. In particular, it achieves better results than Augmented Chebyshev (Chen et al., 2019) and the greedy Highest Cost strategy. The Highest Cost baseline performs poorly because it always selects the most expensive model allowed by the budget, even when cheaper models are sufficient. In contrast, our method can identify these cheaper but effective choices, showing the advantage of using a utility-based objective.

We also study the role of calibration in controlling routing behavior (Right). When using only real-time predictions ($w = 0$), the resulting frontier becomes discontinuous, with large gaps in the middle cost range (between \$0.1 and \$0.7). This means the router often jumps between extreme choices (very cheap or very expensive models), and fails to use mid-range models. Adding the calibration term ($w = 0.2$) smooths this behavior by incorporating historical informa-

*Table 3.* Comparison with lightweight predictors under the same retrieval setup and utility function. AUC is computed over the accuracy–cost Pareto frontier obtained by sweeping $\alpha$; higher is better.

| Method | Predictor | AUC | Best ACC | Min Cost |
|---|---|---|---|---|
| SCOPE | LLM | **0.6986** | **75.00%** | **$0.028** |
| SCOPE$_{\text{NoCoT}}$ | LLM | 0.6608 | 71.03% | $0.029 |
| kNN ($k = 100$) | Non-param. | 0.6422 | 67.46% | $0.168 |
| kNN ($k = 500$) | Non-param. | 0.6379 | 66.27% | $0.175 |
| kNN ($k = 10$) | Non-param. | 0.6301 | 66.27% | $0.183 |
| MLP (Deep) | Low-param. | 0.6261 | 65.87% | $0.193 |
| MLP (Default) | Low-param. | 0.6168 | 67.46% | $0.190 |
| MLP (Small) | Low-param. | 0.6006 | 61.90% | $0.177 |

*Table 4.* Ablation of the two-stage SCOPE training pipeline. SFT mainly improves efficiency by distilling concise rationales, while GRPO drives the main predictive performance gain.

| Model Setting | MAE ↓ | ACC ↑ | Avg. Len. ↓ |
|---|---|---|---|
| Qwen3-4B | 1740 | 59.4% | 2354.9 |
| + SFT | 1754 | 59.0% | 238.7 |
| + SFT & GRPO | **1555** | **77.0%** | **199.1** |

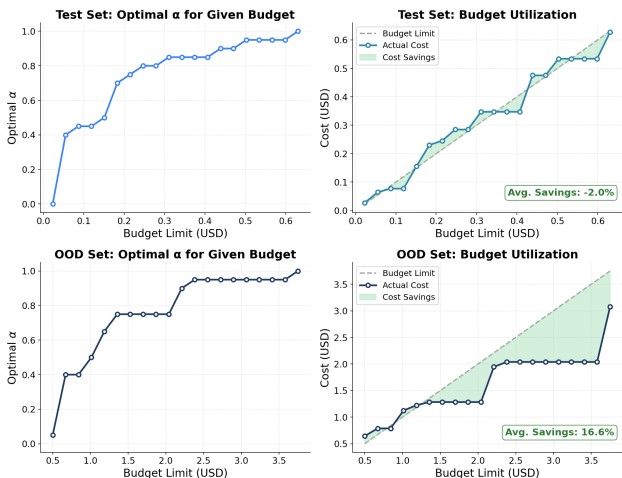

*Figure 8.* **Budget-aware control.** Given a user-specified budget, SCOPE automatically maps it to a trade-off coefficient $\alpha$ and performs routing that closely matches the desired budget. This enables precise, user-controllable accuracy–cost trade-offs at inference time.

tion, leading to a more balanced use of models across different costs. As a result, the system can provide finer control over the accuracy–cost trade-off, instead of being limited to only extreme options.

**Comparison with lightweight predictors.** We further compare SCOPE with lightweight predictors to isolate the role of the reasoning LLM from the retrieved fingerprints and utility-based decision rule. Using the same anchor retrieval and utility function, we implement kNN predictors that vote over retrieved examples and MLP predictors that use query embeddings and aggregate fingerprint statistics. Following Li (2026), we report normalized AUC over the accuracy–cost Pareto frontier obtained by sweeping $\alpha$, where higher AUC indicates better routing quality across budgets.

As shown in Table 3, SCOPE achieves the best Pareto frontier, with a normalized AUC of 0.6986, outperforming both non-parametric kNN predictors and low-parametric MLP predictors. SCOPE also improves over the direct LLM variant without CoT, indicating that explicit reasoning provides additional benefit beyond using an LLM backbone alone. These results suggest that model-performance prediction from retrieved behavioral fingerprints is not merely a feature aggregation problem: the reasoning LLM is better able to interpret semantic similarities, task-specific difficulty, and model-specific behavioral patterns.

**Effect of SFT and GRPO.** We also ablate the two-stage training pipeline to understand whether GRPO is necessary beyond supervised fine-tuning. As shown in Table 4, SFT alone substantially shortens the reasoning traces, reducing

the average reasoning length from 2354.9 tokens to 238.7 tokens. However, it does not improve predictive quality: correctness accuracy remains around 59%, and token MAE remains high. In contrast, adding GRPO improves correctness accuracy to 77.0%, reduces token MAE to 1555, and further shortens reasoning to 199.1 tokens.

This separation suggests that the two stages play different roles. SFT provides an efficient initialization by distilling long hindsight rationales into concise prediction traces, which reduces inference overhead and stabilizes subsequent rollouts. GRPO then optimizes the final prediction objective directly, rather than forcing the model to imitate a specific token-level rationale. In particular, our reward combines a direct correctness reward with an adaptive plateau-with-decay token reward, allowing the model to predict token cost within a useful tolerance region instead of matching exact token counts. This flexible outcome-level optimization is crucial for improving both correctness prediction and token-cost estimation.

### 6.3. Cost Analysis (Q3)

We analyze the cost of SCOPE from three aspects: budget control, token overhead relative to test-time scaling, and the computational cost of domain adaptation.

**Flexible budget control.** SCOPE treats the utility coefficient $\alpha$ as a controllable variable rather than a fixed hyperparameter. Given an incoming query and a user-specified budget limit, we formulate a constrained optimization problem using SCOPE's pre-hoc estimates of accuracy and cost. We then select $\alpha^\star$ that maximizes expected accuracy subject

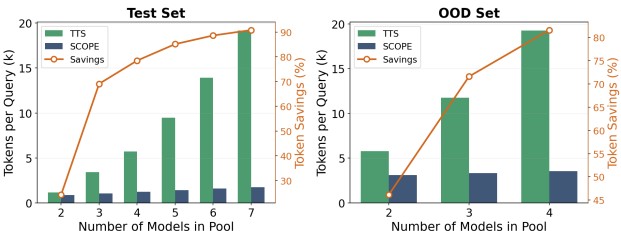

*Figure 9.* **Token comparison between Test-Time Scaling (TTS) and SCOPE.** Green bars show TTS token usage (executing all models per query). Blue bars show SCOPE token usage (pool-wide prediction overhead plus a single model execution). The orange line shows the percentage token savings.

*Table 5.* Wall-clock routing overhead measured on a single 80GB A100 GPU.

| Router / Candidate Pool | Latency per Query |
|---|---|
| MLP Router | $0.048s$ |
| KNN Router | $0.053s$ |
| xRouter | $18.6s$ |
| SCOPE, 3 candidates | $0.31s \pm 0.09s$ |
| SCOPE, 5 candidates | $0.38s \pm 0.09s$ |
| SCOPE, 7 candidates | $0.54s \pm 0.08s$ |
| SCOPE, 9 candidates | $0.89s \pm 0.06s$ |
| SCOPE, 11 candidates | $1.48s \pm 0.09s$ |

to the budget constraint. As illustrated in Fig. 8, varying the budget yields a continuous shift in routing behavior, from cost-sensitive to accuracy-seeking as the constraint relaxes. The full derivation and the finite search procedure are provided in Appendix D.

**Pre-hoc evaluation versus test-time scaling.** We compare SCOPE with standard test-time scaling (TTS), which requires executing multiple candidate models and therefore incurs cost that scales linearly with the pool size. In contrast, SCOPE replaces these executions with lightweight pre-hoc predictions, and executes only one model per query. This reduction relies on hindsight distillation, which shortens the reasoning analysis used for prediction from 2354.9 tokens to 238.7 tokens on average. As shown in Fig. 9, with a pool of 7 models, SCOPE reduces token consumption by 90.8% compared to TTS (1.8k versus 19.2k tokens per query). As the pool grows, the gap widens because SCOPE scales primarily with the distilled prediction tokens, whereas TTS scales with full model generation tokens. A formal scaling analysis is provided in Appendix E.

**Computational cost of domain adaptation.** Compared to baseline routers that require retraining for new domains, SCOPE adapts using anchor-based inference. We quantify this difference using `DeepSeek-V3.2` (37B activated parameters at inference). A representative baseline pipeline executes the 37B model on the full training set (4.8k samples)

to generate labels, and then trains a 4B router, with a total cost of $3.40 \times 10^{18}$ FLOPs. In contrast, SCOPE performs inference only on 250 anchor questions, costing $9.02 \times 10^{16}$ FLOPs. This yields a $38\times$ reduction in compute:

| | **Baseline** | **SCOPE** |
|---|---|---|
| 37B inference samples | 4,778 | 250 |
| 37B inference tokens | 23.3M | 1.22M |
| 4B training tokens | 69.8M | — |
| Total FLOPs | $3.40 \times 10^{18}$ | $9.02 \times 10^{16}$ |
| **Ratio** | **38$\times$** | |

The detailed derivation is provided in Appendix F.

**Wall-clock routing overhead.** Beyond token cost, we further measure the wall-clock overhead introduced by SCOPE. All measurements are conducted on a single 80GB A100 GPU. For the 7-model candidate pool used in our main experiments, SCOPE incurs an average latency of 0.54 seconds per query, with approximately 1,623 input tokens and 199 output tokens. The measured throughput is approximately 20K input tokens and 2.5K output tokens per second.

As shown in Table 5, SCOPE scales smoothly with the number of candidate models because candidate-specific reasoning traces are generated in a single batched forward pass. For typical candidate pools of 5–7 models, the latency remains below 0.6 seconds, and even with 11 candidates the overhead is 1.48 seconds. This is modest compared with the inference time of strong reasoning models on complex queries, which often takes tens of seconds, and can be offset in end-to-end deployment when SCOPE routes queries away from slower and more expensive models.

# 7. Conclusion and Future Work

We introduced SCOPE, a routing framework that decides which model to use by first predicting how accurate and costly each candidate model will be before running it. By relying on models' past behaviors on similar questions, SCOPE can naturally handle new, unseen models without retraining, and gives users direct control over the trade-off between performance and cost. Across a wide range of benchmarks, SCOPE achieves better accuracy–cost trade-offs than individual models, showing that pre-hoc estimation is an effective way to allocate test-time compute.

Beyond single-turn routing, a natural future direction is to extend behavioral fingerprinting to agentic and multi-turn settings. Agent rollouts require step-level model selection, since different models may be better suited for planning, tool use, code generation, verification, or recovery. Trajectory-level fingerprints could support dynamic model assignment across agent workflows, enabling more scalable and cost-aware agent systems.

## Impact Statement

This paper presents work whose goal is to advance the field of machine learning, specifically in the area of efficient and controllable model routing. There are many potential societal consequences of this line of research, including improved accessibility to advanced AI systems through reduced computational cost and better resource allocation. We do not foresee any immediate negative societal impacts arising uniquely from this work beyond those already associated with the general deployment of large language models.

## Acknowledgements

We acknowledge funding support from the National Science Foundation (NSF) under grants IIS-2405974 and IIS-2339216, and from the National Institutes of Health (NIH) under grant R35GM157217.

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

# Appendix

**Appendix Contents.**

## A. More Related Works

Other than model routing, we provide more related works in this section.

**Performance Prediction.** Predicting the solvability of a query is intrinsically linked to Uncertainty Quantification (UQ) and confidence calibration in LLMs. Traditional approaches often rely on white-box signals, such as token probabilities, perplexity, or semantic entropy, to gauge generation quality (Kadavath et al., 2022; Kuhn et al., 2023). While sampling-based techniques like Self-Consistency (Wang et al., 2023b) provide robust confidence estimates, they necessitate multiple expensive inference passes, rendering them impractical for latency-sensitive pre-inference routing. Recent works have explored learning error predictors or quality estimators (Xiao et al., 2025; Khanmohammadi et al., 2025), yet these often depend on internal model states or post-hoc analysis. In contrast, SCOPE operates as a *model-agnostic meta-predictor*. It decouples prediction from execution, estimating both correctness and resource consumption without accessing the target model's logits or triggering its inference, thus enabling purely *ex-ante* decision-making.

**Budget-Aware Inference.** In real-world deployments, latency and cost constraints often necessitate strict adherence to computational budgets. Recent studies on scaling laws indicate that increasing test-time compute can significantly enhance reasoning capabilities, yet this introduces variable and potentially unbounded costs (Snell et al., 2024). To manage these resources, prior works have explored "budget forcing" mechanisms, such as dynamically constraining the reasoning chain length in System 2 models (Han et al., 2025), or optimizing generation hyperparameters (e.g., max tokens) via frameworks like EcoOptiGen to fit constraints (Wang et al., 2023a). However, such methods typically operate *intra-model*, intervening in the decoding process or hyperparameter settings, which may prematurely truncate necessary reasoning steps or degrade output quality. Conversely, SCOPE addresses budget constraints at the *routing level*. By predicting token consumption *ex-ante*, SCOPE enables the construction of a global routing policy that avoids invoking expensive models for queries predicted to exceed token limits, ensuring budget compliance without compromising the integrity of the generation process.

**Representation-based routing.** Recent work has also studied controllable and representation-based model routing. LLM Bandit (Li, 2025) formulates LLM selection as a multi-armed bandit problem with preference-conditioned dynamic routing, allowing users to specify accuracy–cost preferences at inference time. EmbedLLM (Zhuang et al., 2024) learns compact vector representations of LLMs and shows that such representations can capture model characteristics and support downstream routing. Another recent study (Li, 2026) argues that simple kNN-based routers can be competitive with complex learned routers and emphasizes evaluating routing methods across the full accuracy–cost frontier. SCOPE is complementary to these directions but differs in how model information is represented and used. Instead of learning dense model embeddings or relying only on non-parametric neighborhood voting, SCOPE constructs fixed behavioral fingerprints from anchor questions and uses a reasoning predictor to interpret retrieved model behaviors for each target query. This design preserves the adaptability of retrieval-based methods while allowing SCOPE to explicitly predict both correctness and token cost, which are then used for controllable utility-based routing.

*Table 6.* **Engineering Design Principles of SCOPE.** We implement three strategic mechanisms to ensure the system is practical, efficient, and controllable. Empirical validations of these choices are detailed in Section 6.3.

| Goal | Design Choice | Strategic Advantage & Mechanism |
|---|---|---|
| **Generalization** | **Fingerprinting on SCOPE-250** | **No Retraining Needed.** By conditioning on behavioral signatures from a small anchor set (250 queries), SCOPE learns the *skill* of reasoning-based estimation rather than memorizing fixed model IDs, enabling training-free scaling to open-set models. |
| **Efficiency** | **Hindsight Distillation** | **Compressed Reasoning Traces.** We distill concise rationales from teacher models to shorten the Chain-of-Thought length. This significantly reduces token overhead during inference and stabilizes the subsequent GRPO reinforcement learning phase. |
| **Controllability** | **Budget-Constrained $\alpha$-Search** | **Precise Constraint Satisfaction.** Instead of a fixed trade-off, we treat the utility weight $\alpha$ as a dynamic variable. SCOPE solves for the optimal $\alpha$ per query to strictly adhere to the user's monetary budget while maximizing accuracy. |

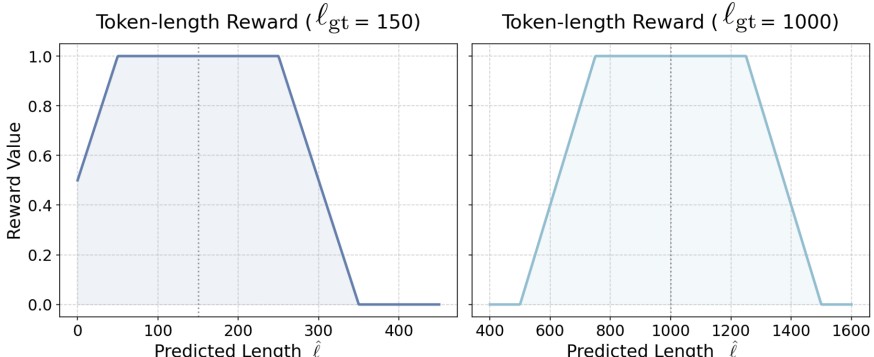

*Figure 10.* Token-length reward. Predictions within $\frac{\tau}{2}$ of the ground truth receive full reward, with linear decay to zero at $\tau$. **Left** $\ell_{gt} = 150, \tau = 200$. **Right** $\ell_{gt} = 1000, \tau = 500$.

# B. Method Details

## B.1. Design Principles

To operationalize these capabilities, we implement three strategic mechanisms outlined in Table 6. We address **generalization** through anchor-based fingerprinting (eliminating the need for retraining), enhance **efficiency** via hindsight distillation (reducing inference overhead), and ensure **controllability** through a budget-constrained optimization solver. These design choices collectively enable SCOPE to scale effectively while maintaining strict adherence to user-defined cost limits.

## B.2. Reward Function Details

In Section 4, we introduced the adaptive token reward $R_{\text{token}}$. Here we provide the exact formulation.

Let $\hat{\ell}$ be the predicted token length and $\ell_{\text{gt}}$ be the ground truth. We define the error distance $d = |\hat{\ell} - \ell_{\text{gt}}|$. To handle the heterogeneity between verbose reasoning models and concise standard models, we define a dynamic tolerance threshold $\tau$:

$$\tau = \max\left(200, \ 0.5 \cdot \ell_{\text{gt}}\right) \tag{9}$$

This threshold implies that for short responses, we allow a fixed buffer of 200 tokens, whereas for long reasoning traces (e.g., 5000 tokens), we allow a relative deviation of 50%.

The reward is shaped using a plateau-with-decay function:

$$R_{\text{token}}(o) = \begin{cases} 1 & \text{if } d \leq \frac{\tau}{2} \\ \frac{\tau - d}{0.5\tau} & \text{if } \frac{\tau}{2} < d \leq \tau \\ 0 & \text{if } d > \tau \end{cases} \tag{10}$$

This shaping encourages the model to land within the "safe zone" ($\leq \tau/2$) while providing a smooth gradient for optimization

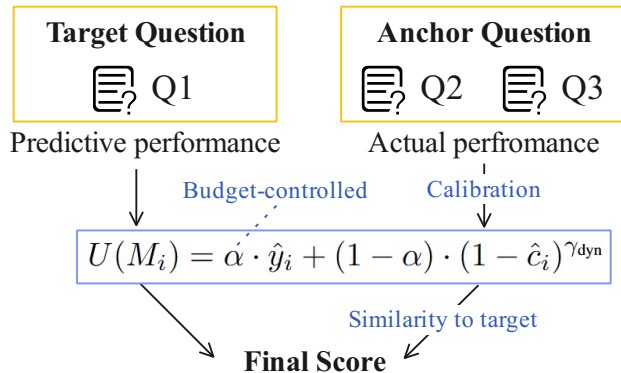

*Figure 11.* **Decision and Calibration Mechanism.** The final routing score aggregates SCOPE's predicted utility (based on estimated correctness $\hat{y}$ and cost $\hat{c}$) with a calibration prior derived from the ground-truth performance of retrieved anchors. This hybrid approach utilizes historical evidence from similar queries to correct potential prediction errors and ensure robust model selection.

when predictions are near the boundary.

## B.3. Detailed Routing Formulation

In this section, we provide the comprehensive mathematical formulation for the utility function and the dynamic calibration mechanism introduced in Section 5.

### B.3.1. COST NORMALIZATION VIA LOG-TRANSFORMATION

Inference costs across the candidate model pool span multiple orders of magnitude (e.g., from \$0.01/M tokens for efficient models to \$10.00+/M tokens for frontier models). A standard linear min-max normalization would compress the differences between efficient models into a negligible range. To address this, we apply a log-transformed min-max normalization to the cost $c$. For a given query cluster in anchor-based calibration, we set $c_{\min}$ and $c_{\max}$ to be the cluster-wise minimum and maximum observed costs. For each input query in online prediction, $c_{\min}$ and $c_{\max}$ here are set to be the per-query minimum and maximum predicted cost among current selected models. The normalized cost $\tilde{c}$ is computed as:

$$\tilde{c} = \frac{\log(c + \epsilon) - \log(c_{\min} + \epsilon)}{\log(c_{\max} + \epsilon) - \log(c_{\min} + \epsilon)} \tag{11}$$

where $\epsilon = 1e^{-6}$ is a small constant for numerical stability. This transformation ensures that the router remains sensitive to price differences even in the low-cost regime.

### B.3.2. DYNAMIC COST SENSITIVITY ($\gamma_{\text{DYN}}$)

The utility function balances accuracy and cost using a trade-off coefficient $\alpha$. However, a linear combination is often insufficient to enforce strict budget constraints. We introduce a dynamic sensitivity factor $\gamma_{\text{dyn}}$ to non-linearly amplify cost penalties when the user prioritizes savings (i.e., low $\alpha$). The utility function is defined as:

$$u = \alpha \cdot \tilde{p} + (1 - \alpha) \cdot (1 - \tilde{c})^{\gamma_{\text{dyn}}} \tag{12}$$

where $\gamma_{\text{dyn}}$ adapts based on $\alpha$:

$$\gamma_{\text{dyn}} = \gamma_{\text{base}} \cdot (1.0 + \beta(1 - \alpha)) \tag{13}$$

In our experiments, we set $\gamma_{\text{base}} = 1.0$ and $\beta = 2.0$. **Intuition:** When $\alpha \to 0$ (cost focus), $\gamma_{\text{dyn}} \to 3.0$. A higher exponent makes the term $(1 - \tilde{c})^{\gamma}$ drop sharply as cost increases, effectively vetoing expensive models unless they guarantee near-perfect accuracy. Conversely, when $\alpha \to 1$, $\gamma_{\text{dyn}} \to 1.0$, allowing for a linear, more tolerant trade-off.

### B.3.3. DYNAMIC CALIBRATION WEIGHT ($w_{\text{CAL}}$)

The final routing score is a weighted ensemble of the model's self-prediction ($u_{\text{pred}}$) and the retrieval-based calibration ($u_{\text{cal}}$). We hypothesize that historical ground truth (anchors) becomes more valuable when maximizing accuracy, whereas predicted

**Category-wise Coverage Analysis (UMAP)**

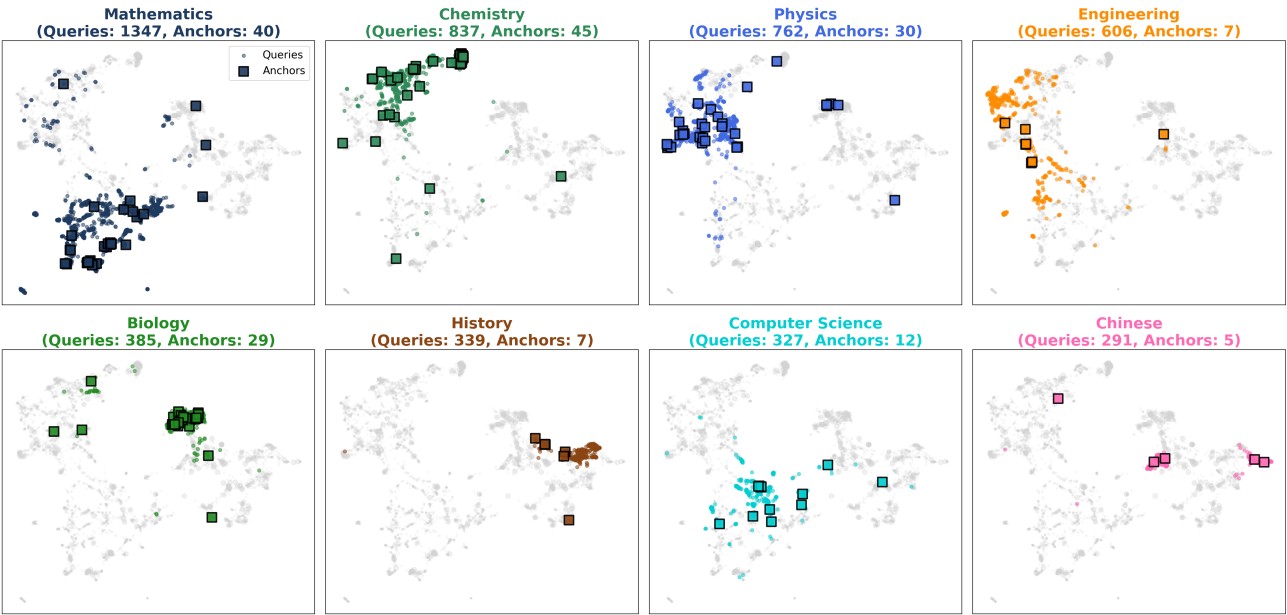

*Figure 12.* **Semantic Coverage Analysis.** We visualize the embedding space of user queries (points) and selected anchors (squares) across 8 major domains using UMAP (McInnes et al., 2020). Despite the high compression ratio (reducing 60k queries to just 250 anchors), the anchors are strategically distributed across the semantic manifold of each category, rather than being clustered in a single region. This spatial coverage confirms that SCOPE-250 provides a representative and diverse basis for retrieval-augmented fingerprinting.

metrics are sufficient for cost control. We define the calibration weight $w_{\text{cal}}$ as:

$$w_{\text{cal}} = w_{\text{base}} \cdot (0.5 + 0.5\alpha) \tag{14}$$

with $w_{\text{base}} = 0.2$. This scales the influence of historical anchors from 10% (when $\alpha = 0$) to 20% (when $\alpha = 1$). The final model selection is determined by:

$$M^* = \arg \max_{M_i \in \mathcal{M}} \left( (1 - w_{\text{cal}}) \cdot u_{\text{pred}}(M_i) + w_{\text{cal}} \cdot u_{\text{cal}}(M_i) \right) \tag{15}$$

This mechanism ensures that for high-stakes queries (high $\alpha$), the router "double-checks" its decision against real historical performance on similar problems.

### B.4. Hyperparameter Sensitivity

SCOPE uses a small number of interpretable routing parameters. The trade-off coefficient $\alpha$ specifies the user's preference between accuracy and cost, and is treated as a controllable variable rather than a fixed hyperparameter. The calibration weight $w$ controls the strength of the historical calibration signal, balancing the anchor-based prior with SCOPE's instance-specific prediction. The dynamic cost-sensitivity parameter $\gamma_{\text{dyn}}$ adjusts the normalized cost term so that it remains comparable with the correctness term in the utility function.

The role of $w$ is analyzed in Fig. 7, where moderate calibration yields the smoothest and most robust Pareto frontier. We further conduct a sensitivity analysis for $\gamma_{\text{dyn}}$, as shown in Table 7.

The normalized AUC varies by less than 0.02 across the tested range, with $\gamma_{\text{dyn}} = 3.0$ performing best. This suggests that SCOPE is not highly sensitive to the exact value of $\gamma_{\text{dyn}}$ and that its routing behavior is governed by a small set of interpretable and robust parameters rather than brittle, setup-specific heuristics.

## C. Routing Effectiveness Details

Table 8 details the diverse portfolio of 11 models employed in our evaluation. The seen (training) pool consists of 7 models spanning various capability and cost tiers, including the high-performance DeepSeek-R1-T2-Chimera (GmbH, 2025) and

*Table 7.* Sensitivity analysis of the dynamic cost-sensitivity parameter $\gamma_{\text{dyn}}$. We report normalized AUC under different values of $\gamma_{\text{dyn}}$.

| $\gamma_{\text{dyn}}$ | Normalized AUC | $\Delta$ AUC |
|---|---|---|
| 2.0 | 0.6884 | $-0.0039$ |
| 3.0 | **0.6923** | ours |
| 4.0 | 0.6852 | $-0.0071$ |
| 5.0 | 0.6784 | $-0.0139$ |

*Table 8.* Model API pricing (USD/1M tokens).

| Model | Act. Params | Input | Output |
|---|---|---|---|
| *Train Set Models (7 models)* | | | |
| DeepSeek-R1-T2-Chimera | ~37B | 0.30 | 1.20 |
| Qwen3-235B-A22B | 22B | 0.18 | 0.54 |
| Nova-2-Lite-V1 | / | 0.30 | 2.50 |
| Qwen3-14B | 14B | 0.05 | 0.22 |
| GPT-OSS-20B | 20B | 0.03 | 0.14 |
| Llama-3.3-70B-Instruct | 70B | 0.10 | 0.32 |
| Gemma-3-27B-IT | 27B | 0.04 | 0.15 |
| *Unseen Models (4 models)* | | | |
| Claude-Sonnet-4.5 | / | 3.00 | 15.00 |
| DeepSeek-V3.2 | ~37B | 0.25 | 0.38 |
| GPT-5-Mini | / | 0.25 | 2.00 |
| Grok-4.1-Fast | / | 0.20 | 0.50 |

the MoE-based Qwen3-235B-A22B (Qwen, 2025). This set also incorporates widely adopted open-weights models such as Llama-3.3-70B-Instruct (Meta, 2024) and Gemma-3-27B-IT (Google, 2025), as well as highly efficient lightweight options including Qwen3-14B (Qwen, 2025), GPT-OSS-20B (OpenAI, 2025a), and Nova-2-Lite-V1 (Amazon, 2025). To rigorously test training-free generalization, we reserve 4 unseen (OOD) models that encompass the latest proprietary advancements: Claude-Sonnet-4.5 (Anthropic, 2025), DeepSeek-V3.2 (DeepSeek, 2025), GPT-5-Mini (OpenAI, 2025b), and Grok-4.1-Fast (xAI, 2025). This selection creates a challenging testbed with pricing varying by two orders of magnitude, from \$0.14 to \$15.00 per million tokens.

Figure 13 provides a direct comparison between SCOPE and each individual model as a deployment baseline. Across these baselines, SCOPE consistently improves Pareto efficiency: when the baseline model is relatively expensive, SCOPE can match or slightly exceed its accuracy while reducing overall cost by shifting easier queries to cheaper models; when the baseline model is relatively cheap, SCOPE increases accuracy by selectively allocating harder queries to stronger models, without incurring the full cost of always using the strongest option. This verifies that the gain is not tied to a particular reference model, but arises from exploiting query-dependent difficulty and the non-dominated structure of the model pool.

Figure 14 visualizes how the induced model portfolio evolves as $\alpha$ varies from cost-oriented routing to accuracy-oriented routing. On the test set, small $\alpha$ concentrates most queries on low-cost models, yielding a low-cost regime with limited accuracy. As $\alpha$ increases, the router gradually mixes in stronger models and reduces reliance on the cheapest options, leading to a smooth increase in set-level accuracy accompanied by a higher total cost. At large $\alpha$, the portfolio becomes dominated by high-capability models, reflecting that the routing objective prioritizes predicted correctness over cost penalties.

A similar but more pronounced trend appears on the OOD set, where the candidate pool contains only unseen proprietary models with a wider price spread. In the cost-focused regime, the portfolio is dominated by the cheapest unseen model, while increasing $\alpha$ introduces more frequent use of higher-capability and higher-price models. The steep cost increase at large $\alpha$ indicates that, under OOD generalization, achieving additional accuracy often requires allocating a non-trivial fraction of queries to frontier models. Overall, Fig. 14 confirms that $\alpha$ functions as an effective and interpretable control knob that induces a coherent shift in routing behavior, rather than causing unstable or erratic portfolio changes.

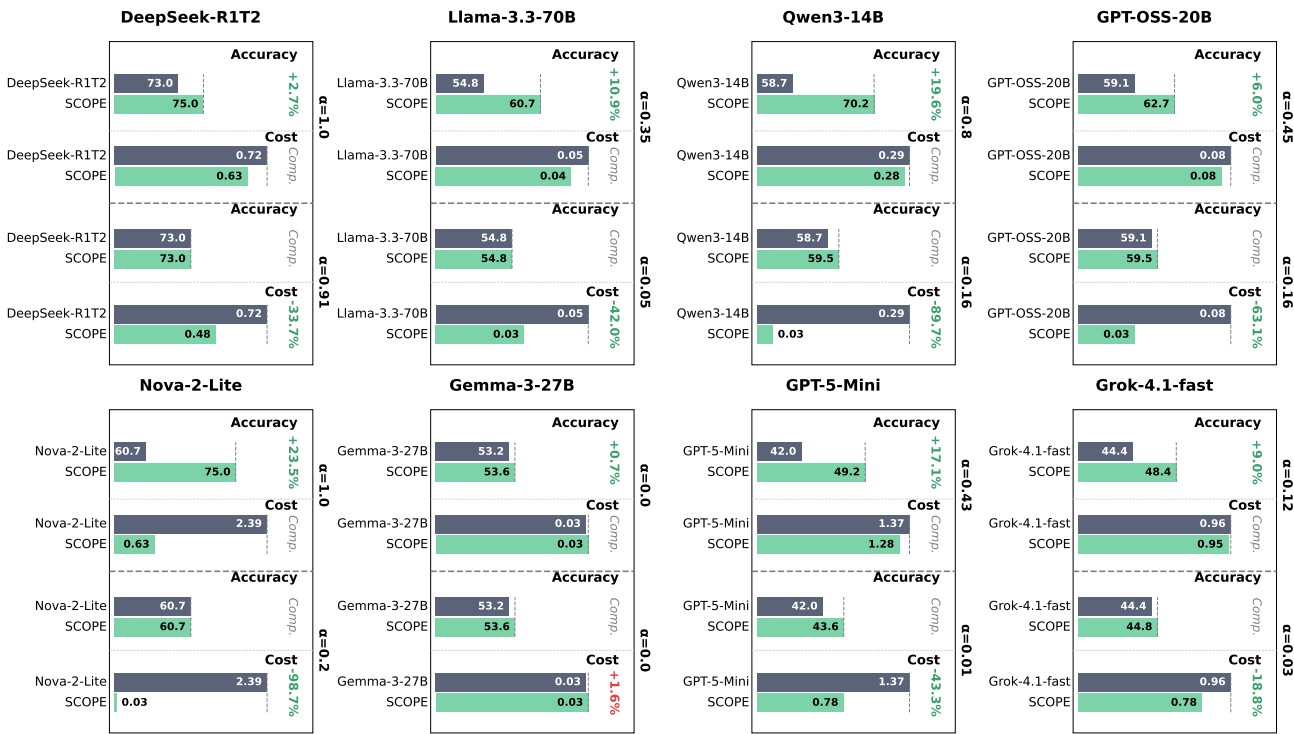

*Figure 13.* **Comparison with Individual Models.** SCOPE achieves strong Pareto efficiency in practice for other eight models. Detailed version of Fig. 4.

### C.1. Confusion Matrix of Oracle and SCOPE Selections

To further analyze routing behavior, we report the confusion matrix between the oracle-selected model and the model selected by SCOPE on the OOD set under $\alpha = 1.0$. Here, the oracle is defined as the cheapest correct model for each query, while the columns indicate the model selected by SCOPE.

*Table 9.* Confusion matrix between the cheapest-correct oracle model and the SCOPE-selected model on the OOD set under $\alpha = 1.0$. Rows denote the oracle-selected model, and columns denote the SCOPE-selected model.

| Oracle Model | SCOPE-selected Model | | | | Total |
|---|---|---|---|---|---|
| | Claude-4.5 | DeepSeek-v3.2 | GPT-5-mini | Grok-4.1-Fast | |
| Claude-4.5 | 7 | 3 | 0 | 1 | 11 |
| DeepSeek-v3.2 | 38 | 27 | 36 | 32 | 133 |
| GPT-5-mini | 13 | 5 | 11 | 5 | 34 |
| Grok-4.1-Fast | 24 | 8 | 23 | 17 | 72 |
| Total | 82 | 43 | 70 | 55 | 250 |

The diagonal entries indicate cases where SCOPE exactly matches the cheapest correct model selected by the oracle. Off-diagonal entries do not necessarily indicate incorrect routing: the selected model may still answer the query correctly, but may not be the cheapest correct option. This distinction is especially important under $\alpha = 1.0$, where SCOPE prioritizes accuracy rather than minimizing cost. The table shows that SCOPE does not collapse to a single dominant model. Instead, its selections are distributed across all four candidate models, suggesting that SCOPE uses the model portfolio adaptively rather than relying on a fixed preferred model.

### C.2. Routing with a Dominant Cost-Effective Model

We further study a natural question: what happens when the candidate pool contains a model that is both highly effective and inexpensive? To simulate this scenario, we add Gemini 3.1 Pro (Google DeepMind, 2026) to the candidate pool. As a

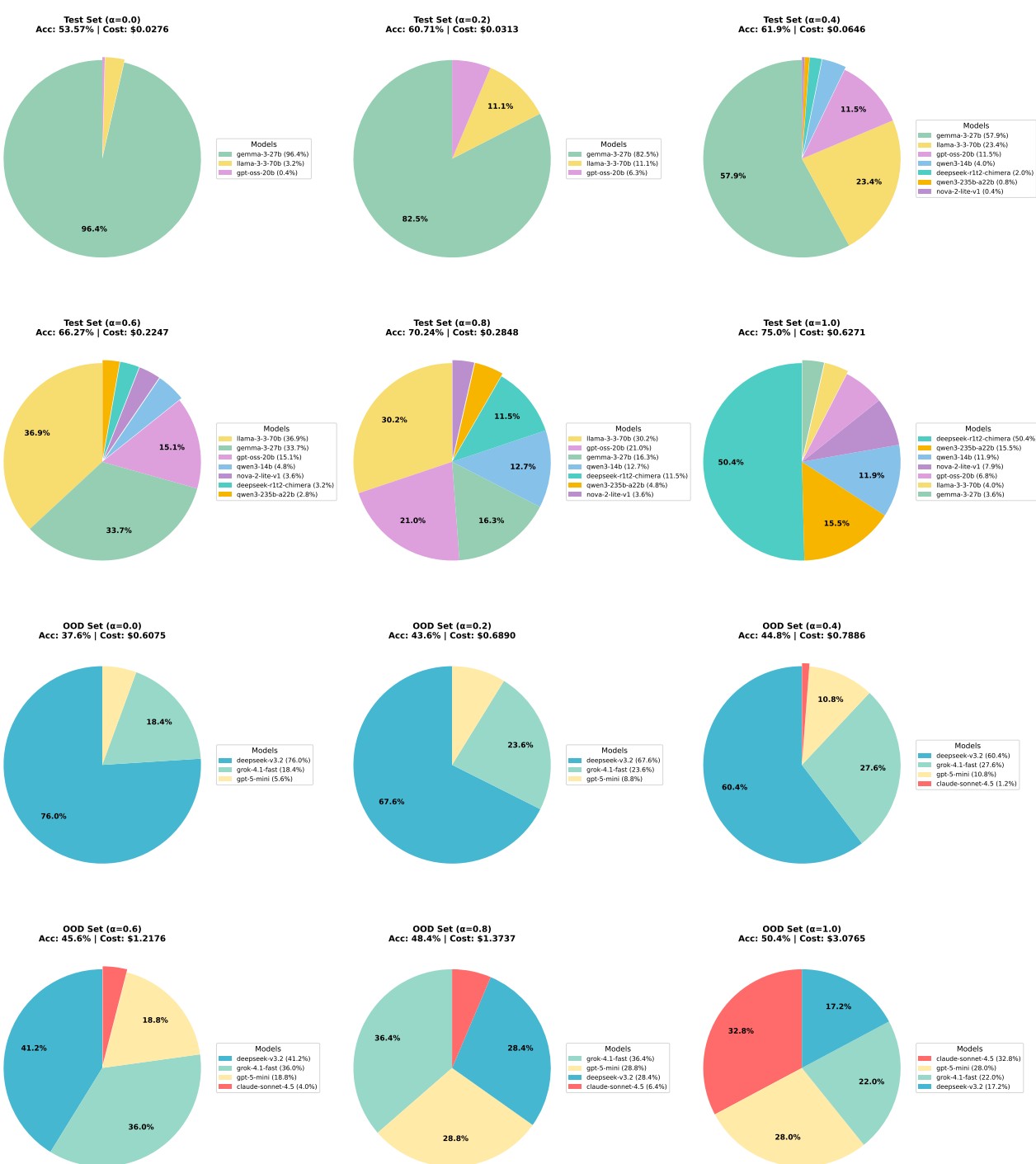

*Figure 14.* **Adaptive Model Portfolio.** All the pie charts from $\alpha = 0$ to $\alpha = 1$. Detailed version of Fig. 5.

standalone model, Gemini 3.1 Pro achieves 65.60% accuracy with a total target-model cost of $22.59 on the evaluation set. We then evaluate SCOPE under three pricing settings. Since Gemini 3.1 Pro is not inexpensive under its original pricing, we additionally simulate two hypothetical pricing regimes by adjusting only its input/output prices while keeping its actual token counts unchanged. This allows us to isolate how SCOPE responds when a strong model becomes increasingly cost-effective.

*Table 10.* SCOPE performance after adding Gemini 3.1 Pro under different simulated pricing regimes. We keep Gemini's actual token counts fixed and vary only its input/output prices.

| Scenario | $\alpha = 0$ | | $\alpha = 0.6$ | | $\alpha = 1$ | |
|---|---|---|---|---|---|---|
| | Acc. | Cost | Acc. | Cost | Acc. | Cost |
| Gemini at original price ($2/$12) | 37.60% | $0.62 | 49.60% | $1.47 | 62.00% | $3.81 |
| Gemini at DeepSeek price ($0.25/$0.38) | 40.80% | $0.64 | 56.00% | $1.18 | 62.00% | $1.96 |
| Gemini near-free ($0.001/$0.001) | 65.60% | $0.00 | 68.40% | $0.47 | 62.00% | $1.69 |

The results show two patterns. First, SCOPE automatically adapts to the cost-performance profile of the available model pool. At Gemini's original pricing, SCOPE reaches 62.00% accuracy at a cost of $3.81 under $\alpha = 1$, reducing cost by approximately 83% compared with always using Gemini 3.1 Pro. As Gemini's simulated price decreases, SCOPE allocates more queries to it and obtains better accuracy–cost trade-offs without retraining or manual reconfiguration.

Second, even when a strong model becomes nearly free, multi-model routing can still provide gains over using that model alone. Under the near-free setting with $\alpha = 0.6$, SCOPE achieves 68.40% accuracy, exceeding the standalone Gemini 3.1 Pro accuracy of 65.60%. This suggests that the candidate models retain complementary strengths: even a strong model is not uniformly optimal across all queries. Therefore, routing remains useful not only for reducing cost, but also for exploiting model complementarity within a heterogeneous model portfolio.

## D. Derivation of Budget-controlled Alpha

We evaluate cost efficiency under the practical setting of model routing. For each query $x$, SCOPE produces *pre-hoc* estimates for every candidate model $M \in \mathcal{M}$, including a predicted correctness $\hat{p}(x, M) \in [0, 1]$ and a predicted inference cost $\hat{c}(x, M) \in \mathbb{R}_+$. Given a trade-off coefficient $\alpha \in [0, 1]$, the router selects a single model by maximizing the utility $U_\alpha$ defined in Sec. 5:

$$\pi_\alpha(x) = \arg\max_{M \in \mathcal{M}} U_\alpha(x, M). \tag{16}$$

**Budgeted choice of $\alpha$ for a query set.** At test time, the system receives a *set* of queries $\mathcal{X} = \{x_1, \ldots, x_n\}$ and a user budget $B$ (USD) for the set. For each query $x \in \mathcal{X}$ and candidate model $M \in \mathcal{M}$, SCOPE outputs pre-hoc estimates $\hat{p}(x, M) \in [0, 1]$ and $\hat{c}(x, M) \in \mathbb{R}_+$. Let $\hat{s}(x, M) \in \mathbb{R}$ denote the cost-related score used inside the utility (e.g., $\hat{s}(x, M) = (1 - \tilde{c}(x, M))^{\gamma_{\mathrm{dyn}}}$ in Sec. 5). For a fixed $\alpha \in [0, 1]$, define the induced routing decision for each query

$$M_\alpha(x) \in \arg\max_{M \in \mathcal{M}} \Big( \alpha\,\hat{p}(x, M) + (1 - \alpha)\,\hat{s}(x, M) \Big). \tag{17}$$

Executing the policy $\pi_\alpha$ on $\mathcal{X}$ yields an *expected* total cost and total accuracy proxy:

$$\widehat{C}(\alpha; \mathcal{X}) = \sum_{x \in \mathcal{X}} \hat{c}\big(x, M_\alpha(x)\big), \tag{18}$$

$$\widehat{P}(\alpha; \mathcal{X}) = \sum_{x \in \mathcal{X}} \hat{p}\big(x, M_\alpha(x)\big). \tag{19}$$

We choose a single $\alpha$ shared across the set by solving the budget-constrained problem

$$\alpha^\star(\mathcal{X}, B) \in \arg\max_{\alpha \in [0,1]} \widehat{P}(\alpha; \mathcal{X}) \quad \text{s.t.} \quad \widehat{C}(\alpha; \mathcal{X}) \leq B. \tag{20}$$

The deployed router then executes $\pi_{\alpha^\star(\mathcal{X}, B)}(x)$ for every $x \in \mathcal{X}$. This formulation matches the deployment requirement: the budget is enforced at the level of the entire workload, while $\alpha$ selects the most accuracy-favorable routing behavior among the budget-feasible trade-offs.

**Finite candidate search for $\alpha^\star(\mathcal{X}, B)$.**  For each query $x \in \mathcal{X}$ and model $M \in \mathcal{M}$, define the affine score

$$
\begin{aligned}
u_{x,M}(\alpha) &= \alpha\,\hat{p}(x, M) + (1 - \alpha)\,\hat{s}(x, M) \\
&= \hat{s}(x, M) + \alpha\big(\hat{p}(x, M) - \hat{s}(x, M)\big).
\end{aligned}
\tag{21}
$$

We assume a deterministic tie-breaking rule is fixed in advance (e.g., choose the lowest-index model), so that $M_\alpha(x)$ in Eq. (17) is a single-valued function of $\alpha$.

For a fixed $x$, the maximizer $M_\alpha(x) \in \arg\max_{M \in \mathcal{M}} u_{x,M}(\alpha)$ can change only at $\alpha$ values where two affine functions intersect. For any distinct $M_i, M_j \in \mathcal{M}$ with $\big(\hat{p}(x, M_i) - \hat{s}(x, M_i)\big) \neq \big(\hat{p}(x, M_j) - \hat{s}(x, M_j)\big)$, define the intersection point

$$
\alpha_{ij}(x) = \frac{\hat{s}(x, M_j) - \hat{s}(x, M_i)}{\big(\hat{p}(x, M_i) - \hat{s}(x, M_i)\big) - \big(\hat{p}(x, M_j) - \hat{s}(x, M_j)\big)}.
\tag{22}
$$

We collect all breakpoints in $(0, 1)$ and add endpoints:

$$
\mathcal{A}(\mathcal{X}) = \{0, 1\} \cup \bigcup_{x \in \mathcal{X}} \{\alpha_{ij}(x) \in (0, 1) : M_i, M_j \in \mathcal{M}\}.
\tag{23}
$$

Let $0 = \tau_0 < \tau_1 < \cdots < \tau_K = 1$ be the sorted unique elements of $\mathcal{A}(\mathcal{X})$. For each interval $(\tau_{k-1}, \tau_k)$, choose an arbitrary representative $\bar{\alpha}_k \in (\tau_{k-1}, \tau_k)$.

**Proposition D.1** (Correctness of the finite search for the set-level budgeted $\alpha$).  *Under the deterministic tie-breaking rule above, for every $k \in \{1, \ldots, K\}$ and every $\alpha \in (\tau_{k-1}, \tau_k)$, the routing decisions satisfy $M_\alpha(x) = M_{\bar{\alpha}_k}(x)$ for all $x \in \mathcal{X}$. Consequently, both $\widehat{C}(\alpha; \mathcal{X})$ and $\widehat{P}(\alpha; \mathcal{X})$ defined in Eqs. (18)–(19) are constant on $(\tau_{k-1}, \tau_k)$. Therefore, there exists an optimal solution to Eq. (20) contained in the finite set*

$$
\{\tau_0, \ldots, \tau_K, \bar{\alpha}_1, \ldots, \bar{\alpha}_K\}.
$$

*Proof.* Fix a query $x \in \mathcal{X}$. For any pair of models $(M_i, M_j)$ with distinct slopes in Eq. (21), the equality $u_{x,M_i}(\alpha) = u_{x,M_j}(\alpha)$ holds at a single point $\alpha = \alpha_{ij}(x)$. All such intersection points in $(0, 1)$ are included in $\mathcal{A}(\mathcal{X})$ by construction. Hence, on any open interval $(\tau_{k-1}, \tau_k)$, the strict ordering of $\{u_{x,M}(\alpha) : M \in \mathcal{M}\}$ cannot change, because no two affine functions intersect inside the interval. With deterministic tie-breaking, the maximizer $M_\alpha(x)$ is therefore constant on $(\tau_{k-1}, \tau_k)$.

Since this holds for every $x \in \mathcal{X}$, the selected model tuple $\{M_\alpha(x)\}_{x \in \mathcal{X}}$ is constant on $(\tau_{k-1}, \tau_k)$. It follows immediately from Eqs. (18)–(19) that $\widehat{C}(\alpha; \mathcal{X})$ and $\widehat{P}(\alpha; \mathcal{X})$ are constant on the same interval. Thus, for each interval, either (i) all $\alpha$ in the interval are infeasible, or (ii) all are feasible and achieve the same objective value. Therefore, an optimal feasible solution can be chosen at an endpoint $\tau_k$ or at a representative $\bar{\alpha}_k$ of a feasible interval. $\square$

## E. Proof for Token Cost Efficiency

**Predictor overhead and comparison to test-time scaling.**  A natural concern is that SCOPE evaluates the entire model pool per query, which could introduce redundant overhead. We make this overhead explicit by writing the *end-to-end* token budget per query. Let $\ell_{\mathrm{pred}}(x, M)$ be the number of tokens consumed by SCOPE when predicting $(\hat{p}, \hat{c})$ for model $M$ on query $x$, and let $\ell_{\mathrm{api}}(x, M)$ be the number of tokens consumed when actually executing $M$ to answer $x$. Under per-query routing, the total token usage is

$$
\ell_{\mathrm{total}}^{\mathrm{SCOPE}}(x) = \sum_{M \in \mathcal{M}} \ell_{\mathrm{pred}}(x, M) + \ell_{\mathrm{api}}\big(x, \pi_\alpha(x)\big).
\tag{24}
$$

Eq. (24) separates the cost into a *prediction overhead* term and a *single* executed-model term. After GRPO training, the predictor is explicitly optimized for concise reasoning and stable formatting, leading to a short average prediction length. Empirically, the average predictor length is 238.7 tokens per model per query, while an untrained baseline predictor (Qwen4B) produces 2354.9 tokens on average, which is a 90% reduction. Therefore, even though SCOPE evaluates all models, the overhead term $\sum_{M \in \mathcal{M}} \ell_{\mathrm{pred}}(x, M)$ remains small relative to the cost of invoking candidate APIs.

This gap becomes more evident when compared with Test-Time Scaling (TTS), which executes multiple models per query and selects the best outcome by observation. If TTS executes a set $\mathcal{S} \subseteq \mathcal{M}$ of models, its total token usage is

$$\ell_{\text{total}}^{\text{TTS}}(x; \mathcal{S}) = \sum_{M \in \mathcal{S}} \ell_{\text{api}}(x, M). \tag{25}$$

In our measurements, executing all available models yields 29,592.9 tokens per query on average, and executing the selected model pool yields 19,174.6 tokens per query on average. In contrast, SCOPE makes *one* API call per query and replaces the remaining calls by lightweight predictions, so the difference between Eq. (25) and Eq. (24) is dominated by the replacement

$$\sum_{M \in \mathcal{S}} \ell_{\text{api}}(x, M) \rightsquigarrow \sum_{M \in \mathcal{M}} \ell_{\text{pred}}(x, M) \; + \; \ell_{\text{api}}\big(x, \pi_\alpha(x)\big),$$

where $\ell_{\text{pred}}$ is orders of magnitude cheaper than $\ell_{\text{api}}$.

Finally, Eq. (24) explains why larger model pools can *strengthen* the cost-efficiency advantage. Both methods scale linearly in the number of considered models, but with different per-model coefficients: TTS scales with the expensive coefficient $\mathbb{E}[\ell_{\text{api}}]$, whereas SCOPE scales with the much smaller coefficient $\mathbb{E}[\ell_{\text{pred}}]$ and adds only one $\mathbb{E}[\ell_{\text{api}}]$ term. As $|\mathcal{M}|$ grows, the token gap between these coefficients amplifies, so the relative efficiency gain of SCOPE over TTS increases.

## F. Proof for Computational Cost Comparison

We derive the computational cost ratio between training a baseline router from scratch versus SCOPE's anchor-based inference for OOD adaptation. We use DeepSeek-V3.2 (37B activated parameters) as a representative example.

### F.1. Baseline Approach: Training a Router

The baseline approach requires two steps:

1. Run the 37B model on the training set to generate supervision signals

2. Train a 4B router on these sequences (prompt + output)

Let:

- $P = 37 \times 10^9$ (activated parameters of the teacher model)

- $P_{\text{router}} = 4 \times 10^9$ (router parameters)

- $N_{\text{tr}} = 4,778$ (training samples)

- $L = L_{\text{ctx}} + L_{\text{gen}} = 208 + 4,665 = 4,873$ tokens per sample

- $E = 3$ (training epochs)

**Step 1: 37B Inference on Training Set.**    Total token throughput:

$$T_{\text{inf}} = N_{\text{tr}} \cdot L = 4778 \times 4873 \approx 23.3 \times 10^6 \text{ tokens}. \tag{26}$$

Inference FLOPs (forward pass only):

$$\begin{aligned} F_{\text{inf}} &= 2 \cdot P \cdot T_{\text{inf}} \\ &= 2 \times 37 \times 10^9 \times 23.3 \times 10^6 \approx 1.72 \times 10^{18} \text{ FLOPs}. \end{aligned} \tag{27}$$

**Step 2: 4B Router Training.**    The training data includes both prompt and model output (the complete sequence):

$$T_{\text{train}} = E \cdot N_{\text{tr}} \cdot L = 3 \times 4778 \times 4873 \approx 69.8 \times 10^6 \text{ tokens}. \tag{28}$$

Training FLOPs (forward + backward + gradient):

$$\begin{aligned} F_{\text{train}} &= 6 \cdot P_{\text{router}} \cdot T_{\text{train}} \\ &= 6 \times 4 \times 10^9 \times 69.8 \times 10^6 \approx 1.68 \times 10^{18} \text{ FLOPs}. \end{aligned} \tag{29}$$

**Total Baseline Cost.**

$$\begin{aligned} F_{\text{baseline}} &= F_{\text{inf}} + F_{\text{train}} \\ &= 1.72 \times 10^{18} + 1.68 \times 10^{18} \approx 3.40 \times 10^{18} \text{ FLOPs.} \end{aligned} \tag{30}$$

The two components are roughly equal: 37B inference accounts for 50.7%, and 4B training accounts for 49.3%.

### F.2. SCOPE Approach: Anchor Inference Only

SCOPE only requires running the 37B model on the anchor set:

- $K = 250$ anchors
- Same $L = 4{,}873$ tokens per sample

Total token throughput:

$$T_{\text{anchor}} = K \cdot L = 250 \times 4873 \approx 1.22 \times 10^6 \text{ tokens.} \tag{31}$$

Inference FLOPs:

$$\begin{aligned} F_{\text{SCOPE}} &= 2 \cdot P \cdot T_{\text{anchor}} \\ &= 2 \times 37 \times 10^9 \times 1.22 \times 10^6 \approx 9.02 \times 10^{16} \text{ FLOPs.} \end{aligned} \tag{32}$$

### F.3. Cost Ratio

$$\frac{F_{\text{baseline}}}{F_{\text{SCOPE}}} = \frac{3.40 \times 10^{18}}{9.02 \times 10^{16}} \approx \mathbf{37.7\times}. \tag{33}$$

**Simplified Formula.** The ratio can be derived analytically:

$$\frac{F_{\text{baseline}}}{F_{\text{SCOPE}}} = \frac{2P \cdot N_{\text{tr}} \cdot L + 6P_{\text{router}} \cdot E \cdot N_{\text{tr}} \cdot L}{2P \cdot K \cdot L} \tag{34}$$

$$= \frac{N_{\text{tr}}}{K} \cdot \left( 1 + \frac{6P_{\text{router}} \cdot E}{2P} \right) \tag{35}$$

$$= \frac{4778}{250} \times \left( 1 + \frac{6 \times 4 \times 3}{2 \times 37} \right) \tag{36}$$

$$= 19.1 \times (1 + 0.97) \tag{37}$$

$$= 19.1 \times 1.97 \approx \mathbf{38\times}. \tag{38}$$

This shows that SCOPE achieves approximately $N_{\text{tr}}/K \approx 19\times$ savings from reduced inference samples, amplified by an additional factor of $\sim 2\times$ since the baseline must also train the router.

### F.4. Memory Comparison

**37B Inference Memory (FP16).**

$$M_{\text{infer}} \approx P \cdot 2\text{B} + M_{\text{KV}} \approx 74\text{GB} + \text{KV cache overhead.} \tag{39}$$

**4B Router Training Memory.**

$$\begin{aligned} M_{\text{train}} &\approx P_{\text{router}} \cdot (2 + 2 + 8)\text{B} + M_{\text{acts}} \\ &\approx 48\text{GB} + 12\text{GB} = 60\text{GB,} \end{aligned} \tag{40}$$

where we account for weights (8GB), gradients (8GB), AdamW optimizer states (32GB), and activations ($\sim$12GB).

Both require high-end GPUs, but SCOPE's anchor inference can be performed via API calls, eliminating local GPU requirements.

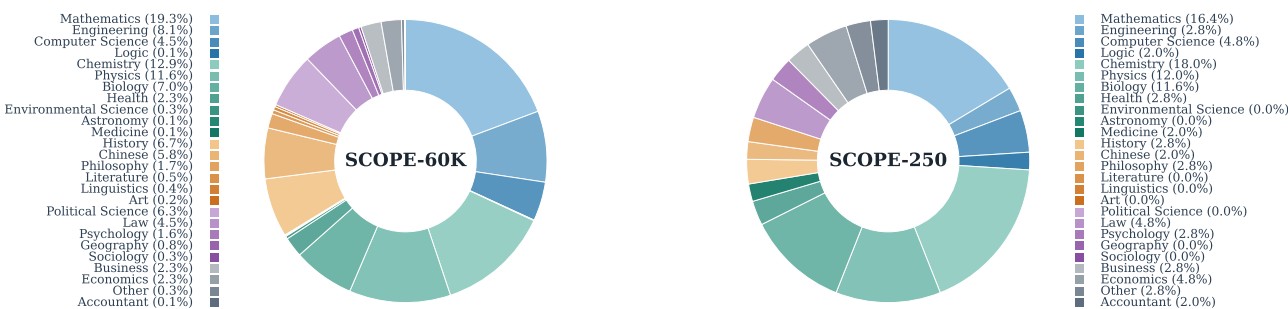

*Figure 15.* **Subject distributions of the SCOPE datasets.** The charts illustrate a broad coverage of disciplines, ranging from STEM fields to Humanities. Notably, the category composition of the SCOPE-250 anchor set closely mirrors that of the full SCOPE-60K dataset. This structural alignment ensures that the compact anchor set serves as a reliable and representative proxy for efficient model fingerprinting and performance estimation.

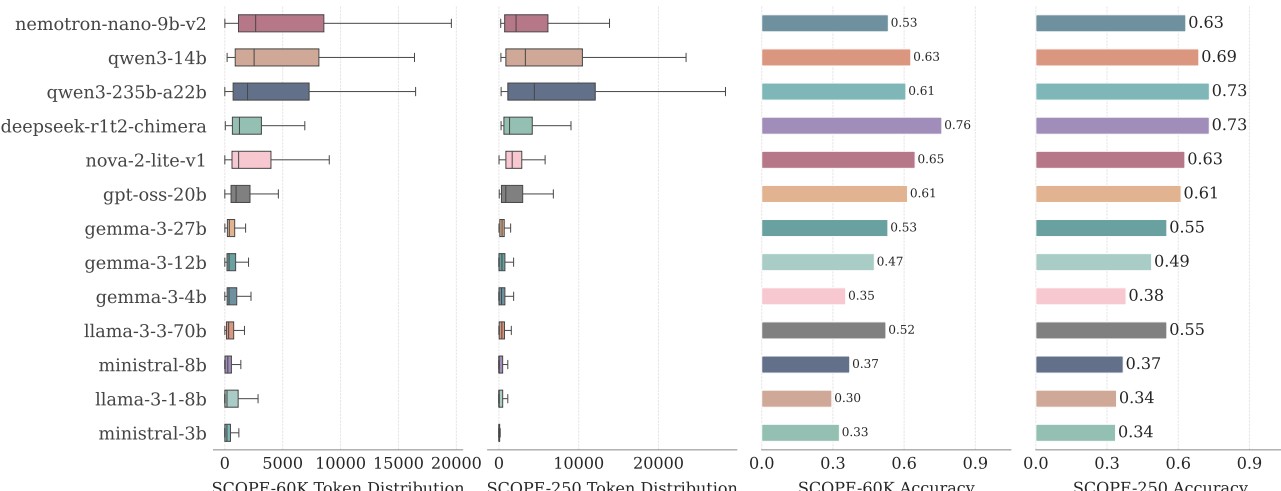

*Figure 16.* **Behavioral diversity and performance comparison across models.** The panels display the token usage distributions (left) and accuracy scores (right) on both the SCOPE-60K and SCOPE-250 datasets. The results highlight the significant distinctions in reasoning patterns (token consumption) and performance capabilities among the evaluated models.

## G. Dataset Details

This section provides empirical evidence supporting the core motivations of SCOPE: the necessity of dynamic routing due to model heterogeneity, and the validity of the anchor-based fingerprinting mechanism. We also specific the details of the dataset we created in this section.

**Heterogeneity in Model Behaviors.** The premise of the SCOPE framework is that models exhibit distinct trade-offs between capabilities and costs, which static routing cannot address. Figure 17 substantiates this by visualizing the accuracy-cost landscape on the SCOPE-60K benchmark. The broad dispersion of data points confirms that no single model dominates across all metrics, validating the need for a controllable outcome estimator. Furthermore, Figure 16 reveals that these differences are intrinsic; each model displays a unique token usage distribution and accuracy profile, effectively serving as a distinguishable "fingerprint" for identification.

**Semantic and Distributional Representativeness of Anchors.** For SCOPE to achieve scalability, the compact anchor set must serve as a high-fidelity proxy for the full distribution. Figure 18 visualizes the semantic space of the SCOPE-250 anchor set via t-SNE, demonstrating a rich semantic coverage that spans diverse domains from STEM to Humanities. This qualitative diversity is quantitatively backed by Figure 15, which confirms that the category composition of the anchor set structurally mirrors that of the full SCOPE-60K dataset.

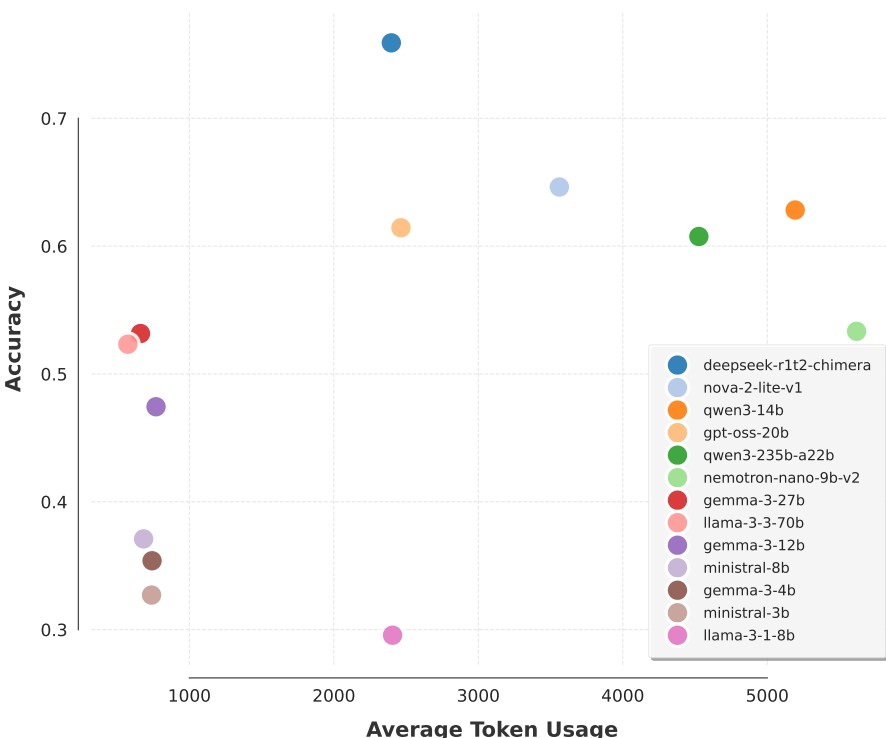

*Figure 17.* **Accuracy versus Average Token Usage on the SCOPE-60K dataset.** The plot illustrates the distinct performance capabilities and varying computational costs associated with each model. The broad scattering of data points highlights the significant diversity in model behaviors and efficiency trade-offs across the SCOPE-60K.

**Behavioral Correspondence.** Finally, we verify that the fingerprints derived from the anchor set serve as effective proxies for global model behaviors. As shown in Figure 16, while the absolute distributions show natural variance due to the sample size difference, the SCOPE-250 anchor set successfully preserves the *relative behavioral characteristics* of the models. Models that exhibit high token consumption or superior accuracy on the full benchmark display comparable tendencies on the anchor set. This trend alignment ensures that the anchor set captures the unique "fingerprint"—the intrinsic capability and cost profile—of each model, providing a discriminative signal for the SCOPE framework to predict performance reliably.

# H. Data Examples

## H.1. SCOPE-60K (Training Set)

**Example metadata.** ID: `rbench_en_001076`   Model: `deepseek-r1t2-chimera`   Category: `Mathematics`

**Ground truth:** B   **Final answer:** B   **Is correct:** True

**Usage:** prompt_tokens: 174, completion_tokens: 2876, total_tokens: 3050

**Question.** Given the system equation:

$$\dot{x} = \begin{bmatrix} \lambda_1 & 1 & 0 \\ 0 & \lambda_1 & 1 \\ 0 & 0 & \lambda_1 \end{bmatrix} x + \begin{bmatrix} a \\ b \\ c \end{bmatrix} u$$

determine $a$, $b$, and $c$ that satisfy the condition for complete state controllability.

Options: A. $c = 0$   B. $c \neq 0$   C. $c \geq 0$   D. $c < 0$   E. $c \leq 0$   F. All other answers are incorrect

**Raw output (truncated).** To determine the condition for complete state controllability, we analyze the controllability matrix $\mathcal{C} = [B\ AB\ A^2 B]$. The last row of $\mathcal{C}$ is $[c, \lambda_1 c, \lambda_1^2 c]$. If $c = 0$, this row becomes all zeros, reducing the rank of $\mathcal{C}$ to at most 2, making

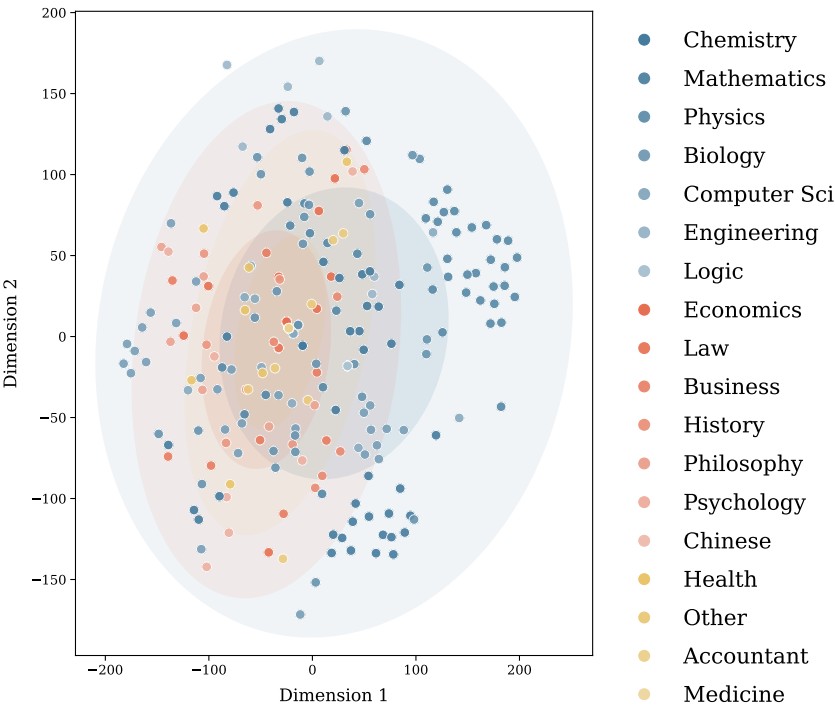

*Figure 18.* **t-SNE of the SCOPE-250 set.** The projection demonstrates the rich semantic composition and broad coverage of the anchor questions, ensuring a comprehensive representation of model behaviors across various domains.

the system uncontrollable. For full rank (rank 3), we require $c \neq 0$. **Final answer: B**

### H.2. SCOPE-250 (Anchor Set)

**Example metadata.** ID: `anchor051`   Model: `deepseek-r1t2-chimera`   Category: `Biology`

**Ground truth:** E   **Final answer:** E   **Is correct:** True

**Usage:** prompt_tokens: 298, completion_tokens: 979, total_tokens: 1277

**Question.** In a group of students, about 36 percent could roll their tongues, a trait determined by a dominant gene. The other 64 percent of the students were nonrollers. Calculate the frequencies of the gene R for tongue rolling and its recessive allele r for nonrolling.

Options: A. $R = 0.5, r = 0.5$   B. $R = 0.7, r = 0.3$   C. $R = 0.4, r = 0.6$   D. $R = 0.18, r = 0.82$   E. $R = 0.2, r = 0.8$
F. $R = 0.6, r = 0.4$   G. $R = 0.3, r = 0.7$   H. $R = 0.64, r = 0.36$   I. $R = 0.36, r = 0.64$   J. $R = 0.8, r = 0.2$

**Raw output (truncated).** Using the Hardy-Weinberg principle, where the frequency of the homozygous recessive genotype (rr) is $q^2$. Given that 64% are nonrollers, $q^2 = 0.64$, so $q = 0.8$. Since $p + q = 1$, we get $p = 0.2$. Verification: Frequency of rollers $= p^2 + 2pq = 0.04 + 0.32 = 0.36$, matching the given 36%. Thus, frequency of R is 0.2 and r is 0.8. **Final answer: E**

### H.3. OOD Test Set

**Example metadata.** ID: `aime_2024_87`   Model: `deepseek-v3.2`   Category: `Math`

**Ground truth:** 699   **Final answer:** 699   **Is correct:** True

**Usage:** prompt_tokens: 92, completion_tokens: 18942, total_tokens: 19034

**Question.** Let $N$ be the greatest four-digit positive integer with the property that whenever one of its digits is changed to 1, the resulting

number is divisible by 7. Let $Q$ and $R$ be the quotient and remainder, respectively, when $N$ is divided by 1000. Find $Q + R$.

**Raw output (truncated).** The only four-digit numbers satisfying the condition are 5624 and 5694, with 5694 being the greatest. Dividing 5694 by 1000 gives quotient $Q = 5$ and remainder $R = 694$. Thus, $Q + R = 5 + 694 = 699$. $\boxed{699}$

## H.4. OOD Anchor Set

**Example metadata.** ID: `anchor051`  Model: `deepseek-v3.2`  Category: `Biology`

**Ground truth:** E  **Final answer:** E  **Is correct:** True

**Usage:** prompt_tokens: 299, completion_tokens: 344, total_tokens: 643

**Question.** In a group of students, about 36 percent could roll their tongues, a trait determined by a dominant gene. The other 64 percent of the students were nonrollers. Calculate the frequencies of the gene R for tongue rolling and its recessive allele r for nonrolling.

Options:  A. $R = 0.5, r = 0.5$  B. $R = 0.7, r = 0.3$  C. $R = 0.4, r = 0.6$  D. $R = 0.18, r = 0.82$  E. $R = 0.2, r = 0.8$
F. $R = 0.6, r = 0.4$  G. $R = 0.3, r = 0.7$  H. $R = 0.64, r = 0.36$  I. $R = 0.36, r = 0.64$  J. $R = 0.8, r = 0.2$

**Raw output (truncated).** The frequency of nonrollers (rr) is 64%, so $q^2 = 0.64$, giving $q = 0.8$ for the recessive allele r. The frequency of the dominant allele R is $p = 1 - q = 0.2$. **Final answer: E**

## H.5. Easy vs. Hard Question Comparison

This subsection illustrates the contrast between easy and hard questions in terms of reasoning complexity and output length.

### Easy Question

**Metadata.** ID: `mmlu_pro_010055`  Category: `Physics`  Completion tokens: **645**

**Ground truth:** H  **Final answer:** H  **Is correct:** True

**Question.** When relatively slow-moving molecules condense from the air, the temperature of the remaining air tends to:

Options: A. condense as well  B. decrease  C. become highly unstable  D. drop rapidly then stabilize  E. fluctuate unpredictably
F. spread out uniformly  G. become extremely cold  H. increase  I. increase then decrease  J. remain unchanged

**Raw output.** When slower-moving molecules condense from the air, they are removed from the gas phase, leaving behind molecules with higher average kinetic energy. Since temperature is a measure of the average kinetic energy of molecules, the remaining air's temperature increases. **Final answer: H**

*Note: The model applies basic thermodynamics reasoning in a straightforward manner.*

### Hard Question

**Metadata.** ID: `gpqa_000247`  Category: `Physics`  Completion tokens: **6444**

**Ground truth:** $10^{62}$ J/K  **Final answer:** $10^{62}$  **Is correct:** True

**Question.** The angular size of the event horizon of a supermassive black hole in the centre of a galaxy at a distance of $d = 10^{10}$ parsecs is measured to be $\theta = 10^{-17}$ degrees. Find the order of magnitude of the entropy of the black hole.

**Raw output (truncated).** The angular size $\theta$ is given as $10^{-17}$ degrees, and distance $d = 10^{10}$ parsecs. Using the small-angle formula, $\theta \approx 2R_s/d$, where $R_s$ is the Schwarzschild radius. Converting $\theta$ to radians ($\theta \approx 1.745 \times 10^{-19}$ rad) and $d$ to meters ($d \approx 3.086 \times 10^{26}$ m), we calculate $R_s \approx 2.69 \times 10^7$ m.

The mass $M$ is derived from $R_s = 2GM/c^2$, yielding $M \approx 1.81 \times 10^{34}$ kg. The event horizon area $A = 4\pi R_s^2 \approx 9.09 \times 10^{15}$ m$^2$.

The entropy $S$ is given by the Bekenstein-Hawking formula:

$$S = \frac{k_B c^3 A}{4\hbar G} \approx 1.20 \times 10^{62}$$

The order of magnitude of the entropy is $10^{62}$. **Final answer:** $10^{62}$

*Note: The model performs multi-step physics calculations involving unit conversions, general relativity, and the Bekenstein-Hawking entropy formula, resulting in $10\times$ more tokens than the easy question.*

### H.6. SCOPE-NoCoT (SFT Inference)

**Example metadata.** ID: `rbench_en_000042`   Model: `deepseek-r1t2-chimera`   Category: `Physics`

**Prompt template (NoCoT).** `### Task`
```
You are a performance prediction expert.
Given a target question, K anchor questions with their performance results,
and a target AI model, predict how the model will perform on the target question,
specifically the output length and correctness after related reasoning analysis.

### Target Model
{model_name}

{anchor_text}

### Target Question
{question}

### Output Format
You may output additional text above, but the FINAL line of your response
MUST follow this exact format:

Predicted Performance: {len: <integer>, correct: <yes|no>}

### Output:
```

**Key feature.** No chain-of-thought is required. The model directly outputs the structured prediction.

### H.7. SCOPE (RL-CoT)

**Example metadata.** ID: `rbench_en_001076`   Model: `deepseek-r1t2-chimera`   Category: `Math`

**Prompt template (CoT).** `### Task`
```
You are a performance prediction expert.
Given a target question, K anchor questions with their performance results,
and a target AI model, predict how the model will perform on the target question,
specifically the output length and correctness after related reasoning analysis.

### Target Model
{model_name}

{anchor_text}

### Target Question
{question}
```

```
### Output Format (STRICT)
Analysis: [Your comprehensive analysis covering anchor patterns,
target question characteristics, and reasoning.]
Predicted Performance: {len: [integer], correct: [yes/no]}

### Output:
```

**Key feature.** Chain-of-thought is required in the Analysis field. The final line remains the same schema.

### H.8. Hindsight Distillation (SFT Training Data)

**Example metadata.** ID: `rbench_en_001076`   Model: `gemma-3-12b`   Category: `Math`

**Anchor serialization (Markdown-style).** `### Anchor Question k`
```
**Question:** {a['question']}
{a['options']}
**Performance:** {len: {a['len']}, correct: {a['correct']}}
```

**Prompt template (Hindsight Distillation).** `### Task`
```
You are a performance prediction expert.
Given a target question, K anchor questions with their performance results,
and a target AI model, predict how the model will perform on the target question,
specifically the output length and correctness after related reasoning analysis.
You MUST provide an analysis before the final prediction.

### Target Model
{model_name}

{anchor_text_markdown}

### Target Question
{question}

### Output Format (STRICT)
Analysis: [Your analysis of anchor patterns and the target question.]
Predicted Performance: {len: [integer], correct: [yes/no]}

### Output:
```

**Key Features:** Hindsight distillation conditions the teacher on realized outcomes (e.g., ground-truth correctness and token cost) to synthesize concise rationales and structured targets. We then distill these teacher-generated outputs to train the student via SFT, while preserving the same output schema as the CoT prompt. The main difference from SCOPE (RL-CoT) is the anchor serialization style (Markdown-like blocks vs. plain text), not the prediction schema.

## I. Test Set Case Studies

### I.1. SCOPE CoT

**Example 1 (TP)**
ID: `rbench_en_000042`
Model: `tngtech/deepseek-r1t2-chimera`
Anchors: `anchor090, anchor092, anchor029, anchor158, anchor138`
Pred: correct=yes, len=3000 tok

GT: correct=yes, len=4603 tok
Conf: 1.00    Match: OK

**Samples (1 total):**

*Sample 1*: correct=yes, len=3000

---

The target question involves calculating the electric field along the z-axis due to a charged ring with a specific charge distribution. The anchor questions cover various electromagnetism topics, including electric fields, polarization, displacement, and charge distributions. The model's performance on these questions varies, with some being answered correctly and others not. The target question is similar in complexity to Anchor Questions 1 and 2, which involved detailed calculations and multiple-choice selections. However, it is more straightforward in terms of the physical setup and mathematical requirements. The model's performance on Anchor Question 3 was incorrect, which involved a simpler concept but required understanding of electrostatics. The target question requires integrating the contributions from different parts of the ring, which might be challenging but not as complex as the previous ones. The model's performance on Anchor Questions 4 and 5 was correct, indicating good handling of advanced topics. Given the model's history of correct answers on similar problems, it is likely to perform well on this question, though there is a risk of error in the integration process or misapplying formulas.
**Predicted Performance: {len: 3000, correct: yes}**

---

**Example 2 (TN)**
ID: `mmlu_pro_011758`
Model: `google/gemma-3-12b-it`
Anchors: `anchor220, anchor086, anchor017, anchor190, anchor137`
Pred: correct=no, len=1000 tok
GT: correct=no, len=646 tok
Conf: 1.00    Match: OK

**Samples (1 total):**

*Sample 1*: correct=no, len=1000

---

The target question involves calculating the dew point and mole fractions of combustion products for propane burning in theoretical air. This is a complex thermodynamic and chemical engineering problem requiring knowledge of stoichiometry, gas behavior, and phase equilibrium. The anchor questions cover various topics including chemical equilibrium, bond energies, reaction conditions, and physics calculations, with varying performance levels. The model's performance on these questions shows mixed results, with some correct answers and others incorrect. The target question is more technical and requires precise calculations, which might be challenging for the model. However, the model has shown capability in handling multi-step problems, especially when the steps are well-defined. The correct answer likely involves determining the mole fractions of each component and then using the ideal gas law to find the dew point. Given the model's past performance on similar complex questions, it might struggle with the exact calculations but could potentially arrive at the correct answer if the steps are followed accurately.
**Predicted Performance: {len: 1000, correct: no}**

---

**Example 3 (TN)**
ID: `rbench_en_000042`
Model: `google/gemma-3-27b-it`
Anchors: `anchor090, anchor092, anchor029, anchor158, anchor138`
Pred: correct=no, len=800 tok
GT: correct=no, len=1243 tok
Conf: 1.00    Match: OK

**Samples (1 total):**

*Sample 1*: correct=no, len=800

---

The target question involves calculating the electric field along the z-axis for a charged ring with a specific charge distribution. The anchor questions cover electromagnetism topics like electric fields, displacement, and charge distributions, but the target question is more about symmetry and integration. The model's performance on these questions was inconsistent, with some being incorrect. However, the target question has a clear mathematical structure and multiple-choice options, which might be easier to handle. The correct answer requires understanding of symmetry and integration of charge contributions. The model might struggle with the integration process but could potentially select the correct option if it recognizes the symmetry.
**Predicted Performance: {len: 800, correct: no}**

---

**Example 4 (TN)**
ID: `rbench_en_000042`
Model: `google/gemma-3-4b-it`
Anchors: `anchor090, anchor092, anchor029, anchor158, anchor138`
Pred: correct=no, len=800 tok
GT: correct=no, len=1118 tok
Conf: 1.00    Match: OK

**Samples (1 total):**

*Sample 1*: correct=no, len=800

---

The target question involves calculating the electric field along the z-axis due to a charged ring with a specific charge distribution. The anchor questions cover electromagnetism topics like electric fields, displacement, and charge distributions, with varying complexity. The model's performance on these questions shows incorrect answers and longer response times, indicating potential difficulties with theoretical physics and multi-step calculations. The target question requires integrating the contributions from different parts of the ring, considering symmetry and charge distribution. The correct answer likely involves recognizing the symmetry of the problem and simplifying the integral, which might be challenging for the model. The options are similar in structure, so careful analysis is needed to select the right one.

**Predicted Performance: {len: 800, correct: no}**

---

**Example 5 (TN)**
ID: `mmlu_pro_011758`
Model: `openai/gpt-oss-20b`
Anchors: `anchor220, anchor086, anchor017, anchor190, anchor137`
Pred: correct=no, len=12000 tok
GT: correct=no, len=3152 tok
Conf: 1.00    Match: OK

**Samples (1 total):**

*Sample 1*: correct=no, len=12000

---

The target question involves calculating the dew point and mole fractions of combustion products for propane burning in dry 125% theoretical air. This is a complex thermodynamic and chemical engineering problem requiring knowledge of stoichiometry, gas laws, and phase equilibrium. The anchor questions cover various topics including chemical equilibrium, bond energy, reaction conditions, isotope analysis, and proton energy conversion. The performance on these questions varies, with some being answered correctly and others not. The model's performance on the target question will depend on its ability to handle multi-step calculations and apply relevant chemical principles. The correct answer requires precise calculations involving stoichiometry, ideal gas law, and dew point determination, which are more advanced than the previous questions. However, the model has shown accuracy in similar multi-step problems (e.g., Anchor Question 2 and 5), suggesting it might perform well. The incorrect answers in Anchor Questions 1 and 4 indicate potential issues with complex chemical equilibrium and isotope-related problems, but the target question is more structured and has clear options, which might aid in prediction.

**Predicted Performance: {len: 12000, correct: no}**

---

**Example 6 (TN)**
ID: `rbench_en_000042`
Model: `meta-llama/llama-3.1-8b-instruct`
Anchors: `anchor090, anchor092, anchor029, anchor158, anchor138`
Pred: correct=no, len=400 tok
GT: correct=no, len=650 tok
Conf: 1.00    Match: OK

**Samples (1 total):**

*Sample 1*: correct=no, len=400

The target question involves calculating the electric field along the z-axis for a charged ring with a specific charge distribution. The anchor questions cover electromagnetism topics like electric fields, displacement, and charge distributions, but the target question is more about symmetry and integration of charge contributions. The model's performance on these questions was incorrect, suggesting possible issues with understanding charge distributions and symmetry. However, the target question has a clear structure with options that can be derived through integration and symmetry arguments. The answer likely involves integrating the contributions from each infinitesimal charge element on the ring, considering the symmetry of the problem. The correct answer is probably F or B, as they have similar forms but differ in coefficients. Given the model's previous incorrect answers, it might struggle with the exact coefficient calculation.

**Predicted Performance: {len: 400, correct: no}**

---

**Example 7 (TP)**
ID: `mmlu_pro_011758`
Model: `meta-llama/llama-3.3-70b-instruct`
Anchors: `anchor220, anchor086, anchor017, anchor190, anchor137`
Pred: correct=yes, len=1200 tok
GT: correct=yes, len=828 tok
Conf: 1.00    Match: OK

**Samples (1 total):**

*Sample 1*: correct=yes, len=1200

The target question involves calculating the dew point and mole fractions of combustion products for propane burning in theoretical air. This is a complex thermodynamic and chemical engineering problem requiring knowledge of stoichiometry, gas laws, and phase equilibrium. The anchor questions cover various topics including chemical equilibrium, bond energies, reaction conditions, isotopic analysis, and heat transfer, with varying performance levels. The model performed well on Anchor Question 3 (correct: yes) but struggled with others. The target question's complexity is similar to Anchors 1, 2, 4, and 5, which had longer responses but mixed correctness. Given the model's past performance on similar technical questions, it might require significant computation and understanding of thermodynamic principles to arrive at the correct answer. The correct answer likely requires precise calculations and familiarity with combustion processes and dew point determination.

**Predicted Performance: {len: 1200, correct: yes}**

---

**Example 8 (TN)**
ID: `mmlu_pro_011758`
Model: `mistralai/ministral-3b`
Anchors: `anchor220, anchor086, anchor017, anchor190, anchor137`
Pred: correct=no, len=1000 tok
GT: correct=no, len=5 tok
Conf: 1.00    Match: OK

**Samples (1 total):**

*Sample 1*: correct=no, len=1000

The target question involves calculating the dew point and mole fractions of combustion products for propane burning in dry 125% theoretical air. This is a complex thermodynamic and chemical engineering problem requiring knowledge of stoichiometry, gas laws, and phase equilibrium. The anchor questions cover various topics including chemical equilibrium, bond energy, reaction conditions, and physics calculations. The model's performance on these questions varies: it struggles with equilibrium and thermodynamics (Anchor 1, 3) but performs better on physics-based problems (Anchor 5). The target question is more technical and may require precise calculations, which the model might handle with some difficulty. However, the presence of multiple choice options could aid in narrowing down the correct answer. The model's previous performance on similar technical questions suggests it might not fully grasp the required concepts, leading to an incorrect answer.

**Predicted Performance: {len: 1000, correct: no}**

---

**Example 9 (TN)**
ID: `rbench_en_000042`
Model: `mistralai/ministral-8b`
Anchors: `anchor090, anchor092, anchor029, anchor158, anchor138`
Pred: correct=no, len=800 tok
GT: correct=no, len=1279 tok
Conf: 1.00    Match: OK

**Samples (1 total):**

*Sample 1*: correct=no, len=800

> The target question involves calculating the electric field along the z-axis for a charged ring with a specific charge distribution. The anchor questions cover electromagnetism topics like electric fields, displacement, and charge calculations, but none directly match this scenario. The model's performance on these anchors was incorrect, suggesting possible issues with understanding complex charge distributions and field calculations. However, the target question is a standard electrostatics problem with a clear mathematical approach. The model might struggle with integrating the contributions from different charge regions and applying symmetry. The answer options are similar in structure, so the model might select based on partial correctness rather than full derivation. Given the model's past performance on similar problems, it may not fully derive the correct expression but could guess among the options.
>
> **Predicted Performance: {len: 800, correct: no}**

**Example 10 (TN)**
ID: `rbench_en_000042`
Model: `nvidia/nemotron-nano-9b-v2`
Anchors: `anchor090, anchor092, anchor029, anchor158, anchor138`
Pred: correct=no, len=3500 tok
GT: correct=no, len=16384 tok
Conf: 1.00    Match: OK

**Samples (1 total):**

*Sample 1*: correct=no, len=3500

> The target question involves calculating the electric field along the z-axis for a charged ring with non-uniform charge distribution. The anchor questions cover electromagnetism topics like electric fields, displacement, and charge distributions. The model performed well on questions requiring detailed calculations (Anchor 1 and 2) but struggled with conceptual questions (Anchor 3 and 4). The target question is similar in complexity to Anchor 1 and 2, involving integration and vector calculus. However, it also requires understanding of charge distribution and symmetry. The model's performance on similar problems suggests it might handle the mathematical aspects well but may struggle with the specific application of charge distribution to find the electric field. The correct answer likely involves integrating the contributions from each segment of the ring, considering the symmetry and the direction of the electric field along the z-axis. The options provided are all in the form of vectors with z-component, which aligns with the expected result.
>
> **Predicted Performance: {len: 3500, correct: no}**

**Example 11 (TP)**
ID: `2010-2022_Math_I_MCQs_91`
Model: `amazon/nova-2-lite-v1`
Anchors: `anchor223, anchor245, anchor241, anchor047, anchor039`
Pred: correct=yes, len=400 tok
GT: correct=yes, len=342 tok
Conf: 1.00    Match: OK

**Samples (1 total):**

*Sample 1*: correct=yes, len=400

> The target question is a complex math problem involving complex numbers, specifically simplifying the expression (3+i)/(1+i). This type of question requires knowledge of complex number arithmetic, including multiplying numerator and denominator by the conjugate of the denominator to simplify. The anchor questions also involve mathematical reasoning, but with varying levels of complexity and domains (e.g., ellipse properties, probability, calculus, tax law, logic). The model's performance on these questions varies, with some being answered correctly and others not. However, the target question is more straightforward in terms of mathematical operations and has a clear step-by-step solution. The model's performance on similar math problems (like Anchor Question 1 and 2) was incorrect, but the target question's structure is more direct and less likely to involve errors in multi-step reasoning. The correct answer can be determined by performing the division of complex numbers using the conjugate method, which is a standard procedure. Given the model's past performance on math problems, it might struggle with the steps but should be able to handle this relatively simple calculation.
>
> **Predicted Performance: {len: 400, correct: yes}**

**Example 12 (TN)**
ID: `rbench_en_000042`

Model: `qwen/qwen3-14b`
Anchors: `anchor090, anchor092, anchor029, anchor158, anchor138`
Pred: correct=no, len=12000 tok
GT: correct=no, len=16384 tok
Conf: 1.00     Match: OK

**Samples (1 total):**

*Sample 1*: correct=no, len=12000

---

The target question involves calculating the electric field along the z-axis for a charged ring with a specific charge distribution. The anchor questions cover electromagnetism topics like electric dipoles, displacement fields, and radiation. The model performed well on complex electromagnetic problems (Anchor 1 and 2) but struggled with conceptual questions (Anchor 3 and 4). The target question requires understanding of symmetry and integration of charge distributions, which are similar to Anchor 1 and 2. The correct answer likely involves integrating the contributions from each segment of the ring, considering the charge density changes. The model's performance on similar problems suggests it can handle this, but there's a risk of error in the integration or symmetry considerations.

**Predicted Performance: {len: 12000, correct: no}**

---

### Example 13 (TP)
ID: `2010-2022_Math_I_MCQs_91`
Model: `qwen/qwen3-235b-a22b`
Anchors: `anchor223, anchor245, anchor241, anchor047, anchor039`
Pred: correct=yes, len=1000 tok
GT: correct=yes, len=768 tok
Conf: 1.00     Match: OK

**Samples (1 total):**

*Sample 1*: correct=yes, len=1000

---

The target question is a complex math problem involving complex numbers, specifically simplifying the expression (3+i)/(1+i). This type of question requires knowledge of complex number arithmetic, including rationalizing the denominator and simplifying the result. The anchor questions also involve mathematical reasoning, but with varying levels of complexity. For example, Anchor Question 1 involves ellipse properties and trigonometry, while Anchor Question 2 is a probability problem with multiple steps. Anchor Question 3 deals with calculus and function derivatives, and Anchor Question 4 and 5 are more about logical reasoning and identifying fallacies. The target question is more straightforward in terms of mathematical operations but still requires careful step-by-step computation. The model's performance on these anchor questions varies, with some being answered correctly and others not. However, the target question's structure is similar to Anchor Question 1 and 3, which involve mathematical manipulation and logical steps. Given that the target question is a standard complex number problem, it is likely that the model can handle it with proper knowledge of complex number arithmetic. The correct answer can be found by multiplying numerator and denominator by the conjugate of the denominator, which is (1-i). This process involves expanding the numerator and denominator, simplifying the expression, and arriving at the correct answer. The model's previous performance on similar mathematical problems suggests that it may have the necessary skills to solve this question correctly.

**Predicted Performance: {len: 1000, correct: yes}**

---

## I.2. SCOPE NoCoT

### Example 1 (TP)
ID: `rbench_en_000042`
Model: `tngtech/deepseek-r1t2-chimera`
Anchors: `anchor090, anchor092, anchor029, anchor158, anchor138`
Pred: correct=yes, len=2060 tok
GT: correct=yes, len=4603 tok
Conf: 1.00     Match: OK

---

**Predicted Performance: {len: 2060, correct: yes}**

---

### Example 2 (TN)
ID: `mmlu_pro_011758`
Model: `google/gemma-3-12b-it`
Anchors: `anchor220, anchor086, anchor017, anchor190, anchor137`

Pred: correct=no, len=1045 tok
GT: correct=no, len=646 tok
Conf: 0.00    Match: OK

Predicted Performance: {len: 1045, correct: no}

---

**Example 3 (TN)**
ID: rbench_en_000042
Model: google/gemma-3-27b-it
Anchors: anchor090, anchor092, anchor029, anchor158, anchor138
Pred: correct=no, len=1133 tok
GT: correct=no, len=1243 tok
Conf: 0.36    Match: OK

Predicted Performance: {len: 1133, correct: no}

---

**Example 4 (TN)**
ID: rbench_en_000042
Model: google/gemma-3-4b-it
Anchors: anchor090, anchor092, anchor029, anchor158, anchor138
Pred: correct=no, len=1408 tok
GT: correct=no, len=1118 tok
Conf: 0.21    Match: OK

Predicted Performance: {len: 1408, correct: no}

---

**Example 5 (TP)**
ID: rbench_en_000042
Model: openai/gpt-oss-20b
Anchors: anchor090, anchor092, anchor029, anchor158, anchor138
Pred: correct=yes, len=1295 tok
GT: correct=yes, len=1596 tok
Conf: 0.74    Match: OK

Predicted Performance: {len: 1295, correct: yes}

---

**Example 6 (TN)**
ID: rbench_en_000042
Model: meta-llama/llama-3.1-8b-instruct
Anchors: anchor090, anchor092, anchor029, anchor158, anchor138
Pred: correct=no, len=5 tok
GT: correct=no, len=650 tok
Conf: 0.34    Match: OK

Predicted Performance: {len: 5, correct: no}

---

**Example 7 (TP)**
ID: rbench_en_000042
Model: meta-llama/llama-3.3-70b-instruct
Anchors: anchor090, anchor092, anchor029, anchor158, anchor138
Pred: correct=yes, len=674 tok
GT: correct=yes, len=929 tok
Conf: 0.64    Match: OK

Predicted Performance: {len: 674, correct: yes}

---

**Example 8 (TN)**
ID: `mmlu_pro_011758`
Model: `mistralai/ministral-3b`
Anchors: `anchor220, anchor086, anchor017, anchor190, anchor137`
Pred: correct=no, len=498 tok
GT: correct=no, len=5 tok
Conf: 0.00    Match: OK

| **Predicted Performance: {len: 498, correct: no}** |
| --- |

**Example 9 (TN)**
ID: `rbench_en_000042`
Model: `mistralai/ministral-8b`
Anchors: `anchor090, anchor092, anchor029, anchor158, anchor138`
Pred: correct=no, len=811 tok
GT: correct=no, len=1279 tok
Conf: 0.34    Match: OK

| **Predicted Performance: {len: 811, correct: no}** |
| --- |

**Example 10 (TN)**
ID: `rbench_en_000042`
Model: `nvidia/nemotron-nano-9b-v2`
Anchors: `anchor090, anchor092, anchor029, anchor158, anchor138`
Pred: correct=no, len=16384 tok
GT: correct=no, len=16384 tok
Conf: 0.00    Match: OK

| **Predicted Performance: {len: 16384, correct: no}** |
| --- |

**Example 11 (TP)**
ID: `rbench_en_000042`
Model: `amazon/nova-2-lite-v1`
Anchors: `anchor090, anchor092, anchor029, anchor158, anchor138`
Pred: correct=yes, len=6082 tok
GT: correct=yes, len=9288 tok
Conf: 1.00    Match: OK

| **Predicted Performance: {len: 6082, correct: yes}** |
| --- |

**Example 12 (TN)**
ID: `rbench_en_000042`
Model: `qwen/qwen3-14b`
Anchors: `anchor090, anchor092, anchor029, anchor158, anchor138`
Pred: correct=no, len=16384 tok
GT: correct=no, len=16384 tok
Conf: 0.00    Match: OK

| **Predicted Performance: {len: 16384, correct: no}** |
| --- |

**Example 13 (TP)**
ID: `2010-2022_Math_I_MCQs_91`
Model: `qwen/qwen3-235b-a22b`
Anchors: `anchor223, anchor245, anchor241, anchor047, anchor039`
Pred: correct=yes, len=1322 tok
GT: correct=yes, len=768 tok
Conf: 1.00    Match: OK

> **Predicted Performance: {len: 1322, correct: yes}**

---

### I.3. Qwen4B Zero-shot NoCoT

**Example 1 (TP)**
ID: `mmlu_pro_011758`
Model: `tngtech/deepseek-r1t2-chimera`
Pred: correct=yes, len=20 tok
GT: correct=yes, len=7848 tok
Conf: 1.00    Match: OK

> **Predicted Performance: {len: 20, correct: yes}**

---

**Example 2 (TP)**
ID: `2010-2022_Math_I_MCQs_91`
Model: `tngtech/deepseek-r1t2-chimera`
Pred: correct=yes, len=30 tok
GT: correct=yes, len=598 tok
Conf: 1.00    Match: OK

> **Predicted Performance: {len: 30, correct: yes}**

---

### I.4. Qwen4B Zero-shot CoT

**Example 1 (TP)**
ID: `rbench_en_000042`
Model: `tngtech/deepseek-r1t2-chimera`
Pred: correct=yes, len=320 tok
GT: correct=yes, len=4603 tok
Conf: 0.88    Match: OK

**Samples (8 total):**

*Sample 1*: correct=yes, len=4

> The target question involves a charged ring with an antisymmetric charge distribution in the $y$-direction: $\lambda = \lambda\_0$ for $y > 0$ and $\lambda$...

*Sample 2*: correct=yes, len=586

> The target question involves calculating the electric field along the $z$-axis due to a charged ring with a non-uniform charge distribution. The charg...

*Sample 3*: correct=yes, len=480

> The target question involves a charged ring with a symmetric charge distribution—positive on the upper half ($y > 0$) and negative on the lower half (...

*Sample 4*: correct=yes, len=620

> The question involves a non-uniform charge distribution with angular dependence. The model correctly identifies that symmetry arguments can be used to...

*Sample 5*: correct=no, len=0

*Sample 6*: correct=yes, len=300

> The target question involves a charged ring with a charge density that is $\lambda = \lambda\_0$ for $y < 0$ and $\lambda = -\lambda\_0$ for $y > 0$, wh...

*Sample 7*: correct=yes, len=250

> The target question involves a symmetric charge distribution ($\lambda = \lambda_0$ for $y < 0$, $-\lambda_0$ for $y > 0$) on a ring, leading to zero electric field along the $z - a$...

*Sample 8*: correct=yes, len=6

> This is a standard problem involving integration over a disk. The model correctly derived the formula using symmetry and integration. The output inclu...

---

**Example 2 (TP)**
ID: `mmlu_pro_011758`
Model: `tngtech/deepseek-r1t2-chimera`
Pred: correct=yes, len=49 tok
GT: correct=yes, len=7848 tok
Conf: 0.62    Match: OK

**Samples (8 total):**

*Sample 1*: correct=yes, len=5

*Sample 2*: correct=yes, len=180

> [Your comprehensive analysis covering anchor patterns, target question characteristics, and reasoning.]

*Sample 3*: correct=yes, len=15

*Sample 4*: correct=no, len=22

> : The target question involves a thermodynamic combustion analysis requiring dew point calculation, which is structurally similar to the anchor questi...

*Sample 5*: correct=yes, len=12

*Sample 6*: correct=no, len=0

> The target question involves propane combustion in dry 125% theoretical air, requiring both the dew point and mole fractions of combustion products (C...

*Sample 7*: correct=no, len=0

> A combustion reaction involves reacting a hydrocarbon with oxygen to produce carbon dioxide and water. The stoichiometry of the reaction must be balan...

*Sample 8*: correct=yes, len=60

> The target question is a complex chemical engineering problem requiring multi-step reasoning involving stoichiometry, vapor-liquid equilibrium, and th...

---

### I.5. Qwen4B K-shot NoCoT

**Example 1 (TP)**
ID: `rbench_en_000042`
Model: `tngtech/deepseek-r1t2-chimera`
Anchors: `anchor090, anchor092, anchor029, anchor158, anchor138`
Pred: correct=yes, len=1342 tok
GT: correct=yes, len=4603 tok
Conf: 1.00    Match: OK

> **Predicted Performance: {len: 1342, correct: yes}**

---

**Example 2 (TP)**
ID: `mmlu_pro_011758`
Model: `tngtech/deepseek-r1t2-chimera`
Anchors: `anchor220, anchor086, anchor017, anchor190, anchor137`
Pred: correct=yes, len=2150 tok
GT: correct=yes, len=7848 tok
Conf: 1.00    Match: OK

> **Predicted Performance: {len: 2150, correct: yes}**

---

## I.6. Qwen4B K-shot CoT

**Example 1 (TP)**
ID: `rbench_en_000042`
Model: `tngtech/deepseek-r1t2-chimera`
Anchors: `anchor090, anchor092, anchor029, anchor158, anchor138`
Pred: correct=yes, len=3340 tok
GT: correct=yes, len=4603 tok
Conf: 0.75    Match: OK

**Samples (8 total):**

*Sample 1*: correct=no, len=3200

> The target question involves calculating the electric field along the z-axis due to a charged ring with a non-uniform charge distribution (positive on...

*Sample 2*: correct=yes, len=5179

> The target question involves calculating the electric field along the z-axis due to a charged ring with a non-uniform charge distribution. The symmetr...

*Sample 3*: correct=yes, len=5179

> The target question involves calculating the electric field along the z-axis due to a charged ring with a non-uniform charge distribution (positive be...

*Sample 4*: correct=yes, len=2830

> The target question involves calculating the electric field along the z-axis due to a charged ring with a non-uniform charge distribution (positive on...

*Sample 5*: correct=yes, len=1129

> The target question involves calculating the electric field along the z-axis due to a charged ring with a non-uniform charge distribution. The ring ha...

*Sample 6*: correct=yes, len=2830

> The target question involves calculating the electric field along the z-axis due to a charged ring with a non-uniform charge distribution (positive on...

*Sample 7*: correct=no, len=1200

> The target question involves calculating the electric field along the z-axis due to a charged ring with asymmetric charge distribution (positive on th...

*Sample 8*: correct=yes, len=5179

> The target question involves calculating the electric field along the z-axis due to a charged ring with a non-uniform charge distribution. The ring ha...

---

**Example 2 (TP)**
ID: `mmlu_pro_011758`
Model: `tngtech/deepseek-r1t2-chimera`
Anchors: `anchor220, anchor086, anchor017, anchor190, anchor137`
Pred: correct=yes, len=3857 tok
GT: correct=yes, len=7848 tok
Conf: 0.75    Match: OK

**Samples (8 total):**

*Sample 1*: correct=yes, len=3756

> The target question involves combustion analysis with a focus on calculating the dew point and mole fractions of products in a non-stoichiometric air ...

*Sample 2*: correct=yes, len=3500

> The target question involves a combustion reaction with theoretical air calculation and requires determining the dew point and mole fractions of combu...

*Sample 3*: correct=yes, len=4000

> The target question involves a combustion reaction with theoretical air calculation and requires determining the dew point and mole fractions of combu...

*Sample 4*: correct=no, len=4200

> The target question involves combustion analysis with a focus on calculating the dew point and mole fractions of combustion products under specific co...

*Sample 5*: correct=yes, len=3800

> The target question involves a combustion reaction with excess air (125% theoretical air), requiring calculation of the dew point and mole fractions o...

*Sample 6*: correct=yes, len=3200

> The target question involves a combustion reaction calculation requiring determination of dew point and mole fractions of products under specific cond...

*Sample 7*: correct=yes, len=4200

> The target question involves a combustion reaction with theoretical air calculation and requires finding the dew point and mole fractions of products....

*Sample 8*: correct=no, len=4200

> The target question involves a combustion reaction with excess air (125% theoretical air) and requires calculation of both the dew point and mole frac...

---

# J. OOD Set Case Studies

## J.1. SCOPE OOD CoT

**Example 1 (TN)**
ID: `aime2025-ii_12`
Model: `anthropic/claude-sonnet-4_5`
Anchors: `anchor243, anchor225, anchor065, anchor242, anchor230`
Pred: correct=no, len=12000 tok
GT: correct=no, len=9952 tok
Conf: 1.00     Match: OK

**Samples (1 total):**

*Sample 1*: correct=no, len=12000

> The target question involves a recursive sequence of rational numbers and requires finding the remainder of the sum of numerator and denominator when expressed in simplest form. This is a complex problem involving pattern recognition and possibly periodicity in the sequence. The examples provided involve various mathematical concepts such as function properties, calculus, and sequence analysis, with varying levels of difficulty and correctness. The first example had a long response and was correct, indicating the model can handle intricate function-related problems. The second example had a longer response but was incorrect, suggesting possible issues with handling more abstract or calculus-based problems. The third example was correct with a shorter response, showing the model can handle straightforward calculations. The fourth and fifth examples were correct with moderate lengths, indicating the model can handle sequence and function analysis effectively. The target question is similar in complexity to the first and fourth examples, which involved sequences and function properties. However, it also includes elements of number theory (finding remainders and simplifying fractions), which might be more challenging. Given the model's performance on similar problems, it is likely to attempt a detailed analysis of the sequence's behavior, possibly identifying patterns or periodicity. The correct answer requires careful computation and understanding of the recursive formula, which could be error-prone. The model might struggle with the algebraic manipulation needed to simplify the expression for $x_{2025}$, but if it can identify a pattern or periodicity, it could potentially solve it correctly.
> **Predicted Performance: {len: 12000, correct: no}**

**Example 2 (TN)**
ID: `aime2025-ii_12`
Model: `deepseek/deepseek-v3_2`
Anchors: `anchor243, anchor225, anchor065, anchor242, anchor230`
Pred: correct=no, len=15000 tok
GT: correct=no, len=10487 tok
Conf: 1.00     Match: OK

**Samples (1 total):**

*Sample 1*: correct=no, len=15000

> The target question involves a recursive sequence of rational numbers and requires finding the remainder of the sum of numerator and denominator when divided by 1000. This is a complex problem that demands identifying patterns or cycles in the sequence. The examples provided include problems with functional properties, calculus, and algebraic manipulations, but none directly relate to recursive sequences or modular arithmetic. However, the performance on these questions varies: some are answered correctly, while others are not. The target question's complexity is comparable to Example 2 and 4, which involve advanced mathematical reasoning and pattern recognition. Given the high difficulty and the need for precise computation, the model might struggle with the exact steps required to find the cycle or simplify the expression. The correct answer likely requires recognizing a periodicity in the sequence, which could be non-trivial to discover. Since the model performed poorly on similar complex problems (e.g., Example 1 and 2), it might not handle this task well, leading to an incorrect answer.
> **Predicted Performance: {len: 15000, correct: no}**

**Example 3 (TN)**
ID: `aime2025-ii_12`
Model: `google/gemini-3-flash-preview`
Anchors: `anchor243, anchor225, anchor065, anchor242, anchor230`
Pred: correct=no, len=15000 tok
GT: correct=no, len=16380 tok
Conf: 1.00     Match: OK

**Samples (1 total):**

*Sample 1*: correct=no, len=15000

---

The target question involves a recursive sequence of rationals with a specific transformation rule, requiring analysis of the sequence's behavior over many iterations to find $x_{2025}$. This is a complex problem that demands identifying patterns or cycles in the sequence. The previous examples involve advanced topics like function properties, calculus, and algebraic manipulations, which also require deep mathematical insight. The performance on these questions varies, with some being solved correctly and others not. The target question is similar in complexity to Example 1 and 4, which involved intricate function transformations and sequence analysis. However, it is more computational and algorithmic in nature, possibly involving finding a cycle or simplifying the recurrence relation. Given the high difficulty and the need for careful step-by-step computation, the model might struggle with the exact computation required for 2025 iterations, but if it can identify a pattern or cycle, it could potentially solve it. The correct answer requires precise calculation and understanding of the recurrence relation, which might be challenging but feasible with systematic analysis.

**Predicted Performance: {len: 15000, correct: no}**

---

**Example 4 (TP)**
ID: `aime_2024_87`
Model: `openai/gpt-5-mini`
Anchors: `anchor003, anchor194, anchor076, anchor174, anchor115`
Pred: correct=yes, len=1500 tok
GT: correct=yes, len=7809 tok
Conf: 1.00    Match: OK

**Samples (1 total):**

*Sample 1*: correct=yes, len=1500

---

The target question is a complex mathematical problem involving number theory and digit manipulation. It requires understanding of divisibility rules, modular arithmetic, and systematic exploration of numbers with specific properties. The examples provided involve various domains: combinatorics, organic chemistry, computer science, and calculus. The performance on these questions varies in length and correctness, but the model consistently answers correctly. The target question's complexity is comparable to Example 2 and 4, which required detailed reasoning and knowledge of specialized fields. However, it is more mathematical and less reliant on domain-specific knowledge. The model's performance on mathematical problems (like Example 1 and 5) shows it can handle such tasks with moderate success. Given the need for systematic analysis and careful application of number theory concepts, the model might face challenges in ensuring correctness, but the structured nature of the problem could aid in finding the right approach.

**Predicted Performance: {len: 1500, correct: yes}**

---

**Example 5 (TP)**
ID: `aime2025-i_0`
Model: `x-ai/grok-4_1-fast`
Anchors: `anchor043, anchor174, anchor003, anchor072, anchor190`
Pred: correct=yes, len=4000 tok
GT: correct=yes, len=1020 tok
Conf: 1.00    Match: OK

**Samples (1 total):**

*Sample 1*: correct=yes, len=4000

---

The target question is a number theory problem involving base conversion and divisibility. It requires understanding how numbers in different bases relate to each other and applying divisibility rules. The examples provided include a mix of computer science, organic chemistry, and mathematics problems, with varying complexity levels. The model's performance on these questions varies significantly in length and correctness. For instance, it performed well on the hexadecimal conversion (Example 4) and the combinatorial problem (Example 3), but struggled with more complex organic chemistry (Example 2) and lengthy mathematical reasoning (Example 5). The target question is mathematical and requires systematic analysis, similar to Example 3 and 4, which the model handled correctly. However, it also involves abstract reasoning and number theory concepts, which might be challenging. Given the model's past performance on similar mathematical problems, it is likely to perform well, but the complexity could affect the length of the response and correctness.

**Predicted Performance: {len: 4000, correct: yes}**

---

## J.2. SCOPE OOD NoCoT

**Example 1 (TN)**
ID: `aime2025-ii_12`
Model: `anthropic/claude-sonnet-4_5`
Anchors: `anchor243, anchor225, anchor065, anchor242, anchor230`
Pred: correct=no, len=16384 tok
GT: correct=no, len=9952 tok
Conf: 0.00    Match: OK

**Predicted Performance: {len: 16384, correct: no}**

---

**Example 2 (TP)**
ID: `aime_2024_87`
Model: `deepseek/deepseek-v3_2`
Anchors: `anchor003, anchor194, anchor076, anchor174, anchor115`
Pred: correct=yes, len=5540 tok
GT: correct=yes, len=18942 tok
Conf: 1.00    Match: OK

**Predicted Performance: {len: 5540, correct: yes}**

---

**Example 3 (TN)**
ID: `aime2025-ii_12`
Model: `google/gemini-3-flash-preview`
Anchors: `anchor243, anchor225, anchor065, anchor242, anchor230`
Pred: correct=no, len=16384 tok
GT: correct=no, len=16380 tok
Conf: 0.00    Match: OK

**Predicted Performance: {len: 16384, correct: no}**

---

**Example 4 (TP)**
ID: `aime_2024_69`
Model: `openai/gpt-5-mini`
Anchors: `anchor245, anchor003, anchor002, anchor118, anchor115`
Pred: correct=yes, len=1102 tok
GT: correct=yes, len=671 tok
Conf: 1.00    Match: OK

**Predicted Performance: {len: 1102, correct: yes}**

---

**Example 5 (TP)**
ID: `aime_2024_75`
Model: `x-ai/grok-4_1-fast`
Anchors: `anchor003, anchor118, anchor002, anchor245, anchor120`
Pred: correct=yes, len=1291 tok
GT: correct=yes, len=2285 tok
Conf: 1.00    Match: OK

**Predicted Performance: {len: 1291, correct: yes}**

---

