# OpenReview forum: "Models Under SCOPE: Scalable and Controllable Routing via Pre-hoc Reasoning"
_ICML.cc/2026/Conference — ICML 2026 regular_

### Official Review · Reviewer_gtLn · 2026-03-11

**Soundness:** 3
**Presentation:** 3
**Significance:** 3
**Originality:** 3
**Overall Recommendation:** 5
**Confidence:** 4

**Summary:**

The paper proposes a model routing framework (named SCOPE) which predicts the cost and performance of models for an input query, learning from past behaviors, and thus can generalized to unseen situation. Rigorous evaluations show the considerable performance benefit of SCOPE, with ablation study quantifying the importance of different design choices.

**Compliance With Llm Reviewing Policy:**

Affirmed.

**Key Questions For Authors:**

Please check my question in the weakness part.

**Limitations:**

No, I think the cost would be an issue that can be discussed here.

**Strengths And Weaknesses:**

Strength:
- The paper is well written and easy to follow.
- The problem is important and timely. The solution could have important practical applications.
- The design choices are well justified. I especially appreciated how the reasoning affects the performance.
- The evaluation is solid, and the results confirm the claims.
- The authors plan to make the trained model weights and training dataset available on Hugging Face, which is much appreciated.

Weakness:

The paper is well written and clearly explains the design choices, and the results support the claim. The only issue I have is the cost of applying such a framework. Section 6.3 provides a cost analysis, showing that it is much less costly than the alternative, but the cost (in terms of FLOPs) is still considerable. Why should we train and use a big LLM model to decide which LLM model to use? Could not we apply some simple (yet smart) heuristic - beside the random heuristic used as a baseline in this paper - to make this decision? For example, instead of training a model, to select M using Q_K in equation (3), we can select a model that provides the best tradeoff between y_i and c_i? How does such a simple heuristic compare with SCOPE?

---

> ### Author Rebuttal · Authors · 2026-03-31
>
> # Response to simple heuristics
>
> We sincerely thank the reviewer for the positive assessment and for raising this important question. We believe the reviewer’s suggestion can be interpreted in **two ways**, and it is important to distinguish them.
>
> First, if the suggestion is to select the model for the current query using the true tradeoff between $y_i$ and $c_i$, then this corresponds to an **oracle policy**, since $y_i$ (whether model $M_i$ answers correctly) is only known after executing that model. Therefore, such a policy is not directly deployable, but it is a meaningful upper bound.
>
> Second, if the suggestion is instead to approximate this tradeoff from historical similar queries in $\mathcal{Q}_K$, then we fully agree this is the right practical comparison. We test such lightweight alternatives, including kNN-based retrieval heuristics and small MLP predictors (not the knn/mlp router in our paper, but heuristic methods from “Rethinking Predictive Modeling for LLM Routing: When Simple kNN Beats Complex Learned Routers,” arXiv:2505.12601).
>
> **Oracle upper bound**
>
> If the reviewer’s suggestion is interpreted as selecting the model using the true $y_i$-$c_i$ tradeoff for the current query, this is an **oracle upper bound** rather than a deployable heuristic. For reference, the table below shows the theoretical oracle performance, defined as selecting the **cheapest correct model per query** (and, if all candidate models are wrong, the cheapest model), on both the ID and OOD settings:
>
> | Setting | Oracle Acc | Oracle Cost | Normalized AUC |
> |:---:|:---:|:---:|:---:|
> | ID test set | 83.33% | $0.086$ | 0.8065 |
> | OOD set| 64.00% | $0.975$ | 0.5996 |
>
> Compared with this oracle upper bound, SCOPE achieves **75.00%** best accuracy on ID (vs. oracle **83.33%**) and a Normalized AUC of **0.6986** (vs. oracle **0.8065**), recovering **86.6%** of the oracle AUC. This shows that SCOPE closes a large fraction of the gap between simple heuristic baselines and the theoretical optimum, while remaining fully deployable.
>
>
> **Practical lightweight heuristics vs. SCOPE**
>
> The table below compares all methods on the ID test set using Normalized AUC (higher is better, reflecting a better accuracy-cost tradeoff):
>
> - **kNN:** predicts correctness via majority voting and token cost via average length from the top-$K$ retrieved training questions.
> - **MLP:** takes the target query embedding together with statistical summaries of retrieved anchors as input.
>
> | Rank | Method | Predictor Type | Normalized AUC | Best Acc | Min Cost |
> |:---:|---|---|:---:|:---:|:---:|
> | 1 | **SCOPE (Ours)** | LLM + CoT | **0.6986** | **75.00%** | **$0.028$** |
> | 2 | SCOPE$_{\text{NoCoT}}$ | LLM (Direct) | 0.6608 | 71.03% | $0.029$ |
> | 3 | kNN ($k=100$) | Non-parametric | 0.6422 | 67.46% | $0.168$ |
> | 4 | kNN ($k=500$) | Non-parametric | 0.6379 | 66.27% | $0.175$ |
> | 5 | kNN ($k=10$) | Non-parametric | 0.6301 | 66.27% | $0.183$ |
> | 6 | MLP (Deep) | Low-parametric | 0.6261 | 65.87% | $0.193$ |
> | 7 | MLP (Default) | Low-parametric | 0.6168 | 67.46% | $0.190$ |
> | 8 | MLP (Small) | Low-parametric | 0.6006 | 61.90% | $0.177$ |
>
> These results show that simple non-LLM heuristics are indeed viable baselines, but they remain clearly inferior to SCOPE. Our full model achieves a Normalized AUC of **0.6986**, compared with **0.6422** for the best kNN variant and **0.6261** for the best MLP variant, corresponding to relative gains of **8.8%** and **11.6%**, respectively. Moreover, SCOPE reaches a substantially higher best accuracy (**75.00%** vs. **67.46%**) while also operating at a much lower minimum-cost point on the Pareto frontier.
>
> We believe this gap arises because lightweight heuristics mostly rely on local statistical extrapolation from retrieved neighbors, whereas SCOPE performs **query-conditioned reasoning** about why a particular model is likely to succeed or fail on the target query. This is also supported by the gap between SCOPE and SCOPE$_{\text{NoCoT}}$ (**0.6986** vs. **0.6608**), which suggests that the gain does not come merely from using a larger predictor, but from enabling explicit reasoning during pre-hoc estimation.

---

> > ### Author Rebuttal · Reviewer_gtLn · 2026-04-07
> >
> > I thank the authors for the reply and the new results, and encourage them to include these in the paper. Given the reply provided to the other reviewers, I believe my score correctly reflects the contribution, and I therefore retain it.

---

### Official Review · Reviewer_ZN8z · 2026-03-12

**Soundness:** 3
**Presentation:** 3
**Significance:** 3
**Originality:** 2
**Overall Recommendation:** 4
**Confidence:** 4

**Summary:**

SCOPE is a model-routing framework that, instead of selecting from a fixed set of models, predicts how accurate and how expensive each candidate model will be on a given query, then chooses the model that maximizes a user-controlled accuracy–cost trade-off. It has three main pieces: (1) **Model fingerprinting**: for an anchor set of representative queries, each model’s (correctness, token cost) is precomputed; for a new model, one forward pass over the anchor set yields a fingerprint, so the system can handle unseen models without retraining. (2) **Retrieval-augmented prediction**: for a target query, the top-K similar anchors are retrieved and the corresponding fingerprint slice is serialized into a prompt; a reasoning LLM predicts correctness and token cost with chain-of-thought. (3) **Utility-based decision**: a calibrated utility combining predicted performance and anchor-based calibration is maximized over candidates, with a user-specified coefficient \alpha controlling the accuracy–cost trade-off. Experiments on 11 models (7 seen, 4 unseen), SCOPE-60K training data, and SCOPE-250 anchors show that SCOPE can improve accuracy by up to 25.7% when \alpha is high or cut cost by up to 95.1% when \alpha is low, and it generalizes to unseen models (e.g. Claude-Sonnet-4.5) without retraining.

**Compliance With Llm Reviewing Policy:**

Affirmed.

**Final Justification:**

The rebuttal has addressed all my questions.

**Key Questions For Authors:**

- Would you consider adding AUC under the accuracy–cost curve (sweeping \alpha) to make router comparison less dependent on the choice of \alpha?
- Have you tried a simple predictor (e.g. linear or MLP over query + fingerprint embeddings) instead of the reasoning LLM, and if so, how does it compare on the same retrieval setup?
- Could you report results for SFT-only (no GRPO) on the same benchmarks to show how much the RL stage contributes?

**Limitations:**

The paper has no dedicated Limitations section. The Impact Statement mentions positive societal consequences (e.g. improved accessibility through reduced cost and better resource allocation) and states that the authors do not foresee immediate negative impacts unique to this work beyond those of general LLM deployment, but it does not discuss ways routing could have negative effects (e.g. systematic differences in who or which queries get sent to cheaper vs. stronger models, or reinforcement of existing disparities in access to quality).

**Strengths And Weaknesses:**

## Strength

- The fingerprint idea is appealing. Recording how each model behaves on a fixed anchor set and then retrieving similar anchors for a new query gives a concrete, training-free way to adapt to new models—you only need to run the new model on the anchor set once.

- The separation between “predict performance and cost” and “then decide by utility” also makes the accuracy–cost trade-off controllable at inference time via \alpha, which is useful in practice.

- The comparison with individual models and the Pareto-frontier plots show that SCOPE can dominate single-model operating points. The OOD evaluation on unseen models (including Claude-Sonnet-4.5) and the analysis of how the selected portfolio shifts with \alpha (Fig. 5) support the claims about generalization and controllability. The hindsight-distillation setup to keep CoT short (238.7 vs 2354.9 tokens) and the cost comparison with test-time scaling (e.g. 90.8% token reduction) are practical and clearly presented.

## Weakness

- **Related work on model representation and routing.** The model-fingerprint design—using behavioral summaries (e.g. performance on representative queries) rather than model identity—overlaps conceptually with work that learns or uses compact representations of LLMs for routing. Two relevant lines are not cited: (1) **LLM Bandit** (Li, 2025; [arXiv:2502.02743](https://arxiv.org/abs/2502.02743)), which formulates LLM selection as a multi-armed bandit with preference-conditioned dynamic routing and user-specified accuracy–cost trade-offs; (2) **EmbedLLM** (Zhuang et al., 2024; [arXiv:2410.02223](https://arxiv.org/abs/2410.02223)), which learns compact vector representations of LLMs for downstream routing and shows that such representations capture model characteristics and support routing. Citing and contrasting these (e.g. fingerprint vs. learned embeddings, fixed anchors vs. learned representations) would better position SCOPE and clarify the novelty of the behavioral-fingerprint approach.

- **Evaluation metric for accuracy–cost trade-off.** The paper reports accuracy and cost at a few fixed \alpha values (e.g. \alpha = 0, 0.6, 1.0) and uses PGR, average accuracy, and total cost in Table 1. Recent work on LLM routing has argued that reporting performance at a single operating point (or a few points) can be misleading, because different methods may excel at different points on the accuracy–cost curve. In particular, **Li (2025)** (“Rethinking Predictive Modeling for LLM Routing: When Simple kNN Beats Complex Learned Routers,” [arXiv:2505.12601](https://arxiv.org/abs/2505.12601)) advocates evaluating routers by **AUC under the accuracy–cost curve when sweeping the trade-off parameter** (e.g. \alpha), so that the entire Pareto frontier is summarized in a single, comparable metric. I would encourage the authors to add this: sweep \alpha (or the equivalent budget/utility parameter), plot or tabulate accuracy vs. cost, and report AUC (or an analogous aggregate). That would make comparisons with baselines and with future work more rigorous and less sensitive to the choice of \alpha.

- **Is an LLM necessary for the predictor?** SCOPE uses a fine-tuned reasoning LLM (Qwen3-4B with CoT) to predict correctness and token cost from the retrieved fingerprint. The ablations show that CoT and training help over the base model, but there is no comparison to **simpler predictors** that use the same fingerprint (or the same retrieval setup) with a lightweight model. For example, one could embed the query and the fingerprint (or aggregate statistics from the retrieved anchors) and use a **linear model or small MLP** to regress/classify correctness and cost. EmbedLLM and the kNN-based routing in Li (2025) suggest that simple, non-parametric or low-parametric methods can be competitive with complex learned routers in many settings. Adding a baseline such as “linear (or MLP) over query embedding + fingerprint features” would help show whether the full LLM-based reasoning is necessary or whether most of the gain comes from the fingerprint and the utility-based decision. If the simple baseline is close, that would be an important finding for deployability and cost.

- **Is full SFT + RL necessary?** The two-stage pipeline (SFT with hindsight distillation, then GRPO) is heavy. The paper shows that trained SCOPE beats untuned Qwen-4B and that the RL-enhanced CoT variant beats SCOPENoCoT, but it does not ablate **SFT-only vs. SFT+GRPO**. For practitioners, a natural question is whether GRPO is essential or whether SFT alone (possibly with the same hindsight CoT data) would get most of the benefit at lower training cost. A direct comparison (SFT-only vs. SFT+GRPO) on the same Pareto frontier or the same AUC metric would clarify how much the RL stage adds and whether a lighter pipeline is viable.

---

> ### Author Rebuttal · Authors · 2026-03-31
>
> # Response to related work on model representation and routing
> We thank the reviewer for pointing out these highly relevant papers. We will cite both and expand our "Related Work" section to discuss them in the revised version. We agree contrasting these works better positions SCOPE:
>
> **1. Text vs. Embeddings (Representation):** While EmbedLLM and LLM Bandit compress model behaviors into vector embeddings, SCOPE keeps them as plain text. This is because our router is a reasoning model. By reading the actual text of past behaviors, SCOPE treats performance prediction as a reasoning problem, which scales to unseen models without retraining.
>
> **2. 250 Fixed Anchors (Efficiency):** While all these methods use past data to guide routing, SCOPE specifically compresses this history into a carefully designed set of exactly 250 "anchor" questions. This tiny anchor set massively reduces the overhead of adding a new model. The only cost to support a newly released model is simply running those 250 questions.
>
> # Response to AUC metric
> We thank the reviewer. We agree that evaluating the entire Pareto frontier via AUC is more rigorous than comparing isolated $\alpha$ points.
>
> Following your suggestion, we have incorporated the AUC metric (normalized so the maximum accuracy is 100 and the maximum cost is 1) as advocated in Li (2025), which we will add the metric and cite the paper in the revised version.
>
> To provide a comprehensive analysis, we used this AUC metric to map the full accuracy-cost frontier and compare SCOPE directly against baseline embedding-based predictors. **Please refer to our next response (regarding "other predictors") for the detailed AUC results and discussion.**
>
> # Response to other predictors
>
> We thank the reviewer for this excellent suggestion. As promised in our previous response, we mapped the full accuracy-cost Pareto frontier using the AUC metric advocated in Li (2025).
>
> To answer whether an LLM is necessary, we implemented the exact simple predictors you suggested using the **same** anchor retrieval and utility function, isolating the impact of the predictor itself:
>
> - **kNN:** Uses majority voting for correctness and average token length from the top-K retrieved questions in the training set.
> - **MLP:** Takes the target query embedding and statistical summaries of the retrieved anchors as input.
>
> **Test Set Pareto Frontier AUC Results:**
> | Rank | Method | Predictor Type | Normalized AUC | Best Acc   | Min Cost   |
> | - | - | - | - | - | - |
> | 1 | **SCOPE (Ours)**  | LLM + CoT | **69.86** | **75.00%** | **$0.028** |
> | 2 | **SCOPE_NoCoT**   | LLM (Direct)   | 66.08 | 71.03% | $0.029 |
> | 3 | **kNN (k=100)**   | Non-parametric | 64.22 | 67.46% | $0.168 |
> | 4 | **kNN (k=500)**   | Non-parametric | 63.79 | 66.27% | $0.175 |
> | 5 | **kNN (k=10)**    | Non-parametric | 63.01 | 66.27% | $0.183 |
> | 6 | **MLP (Deep)**    | Low-parametric | 62.61 | 65.87% | $0.193 |
> | 7 | **MLP (Default)** | Low-parametric | 61.68 | 67.46% | $0.190 |
> | 8 | **MLP (Small)**   | Low-parametric | 60.06 | 61.90% | $0.177|
>
> The results demonstrate that utilizing an LLM predictor is highly effective and beneficial for this task. Translating past behavior into future performance prediction naturally aligns with a logical deduction process. Leveraging an LLM's semantic understanding significantly improves prediction quality, and explicitly enforcing a reasoning step (CoT) further boosts the accuracy-cost trade-off (0.6986 vs. 0.6608 AUC).
>
> # Response to SFT + RL
> We thank the reviewer. We agree that a lighter pipeline is always desirable. However, our ablation below shows that both stages serve distinct and indispensable purposes: SFT provides vital efficiency, while GRPO (RL) drives the actual performance gain.
>
> **SFT vs. SFT+GRPO Ablation Results:**
>
> | Model Setting | Token MAE | Correctness ACC | Avg. Reasoning Length |
> |-| -| -| -|
> | Base Qwen3-4B | 1740 | 59.4% | 2354.9 |
> | **+ SFT Only** | 1754 | 59.0% | 238.7 |
> | **+ SFT & GRPO** | **1555** | **77.0%** | **199.1** |
>
> Standard supervised fine-tuning (SFT) strictly penalizes deviations from exact ground-truth reasoning trajectories. However, predicting the exact token count of another model is both unnecessary and nearly impossible. Also, forcing a model to rigidly imitate specific long reasoning steps distracts it from its primary objective—predicting accuracy. Consequently, this rigid, token-level objective yields suboptimal correctness and a high length prediction error.
>
> RL naturally resolves this by optimizing directly for the final outcome rather than exact trajectory matching. We decouple the objective by introducing a direct, verifiable reward for routing correctness alongside a custom "plateau-with-decay" reward for token length. This provides an adaptive token tolerance, rewarding the model for landing in a "safe zone" of token estimates. By freeing the model from rigid imitation, this flexible optimization boosts correctness to 77.0% and lowers the MAE to 1555.

---

> > ### Author Rebuttal · Reviewer_ZN8z · 2026-04-02
> >
> > Thanks for addressing all my questions. I have raised my score.

---

> > > ### Author Response · Authors · 2026-04-02
> > >
> > > Dear Reviewer,
> > >
> > > Thank you very much for your thoughtful feedback and for recognizing the improvements reflected in our rebuttal. We truly appreciate your time and consideration, and we are grateful for your decision to increase the score.
> > >
> > > We will incorporate the clarifications and improvements discussed in our rebuttal into the revised version of the manuscript.
> > >
> > > Thank you again for your constructive feedback.

---

### Official Review · Reviewer_1G5H · 2026-03-12

**Soundness:** 2
**Presentation:** 3
**Significance:** 2
**Originality:** 3
**Overall Recommendation:** 4
**Confidence:** 2

**Summary:**

This paper proposes SCOPE, a RL-based model to predict the accuracy and token costs of different LLMs for a given query for optimal tradeoff. The model starts by retrieving similar queries from anchor datasets and constructing concatenated prompts, then it leverages CoT to predict the utility trained by SFT and GRPO. The final model selection is based on the utility and historical calibration from similar anchor queries. Experiments demonstrate better performance on multiple seen and unseen models

**Compliance With Llm Reviewing Policy:**

Affirmed.

**Final Justification:**

Q3 in the latest author response seems to remain a concern. The expected baseline would be the cheapest Gemma alone rather than deepseek, which helps to evaluate how much SCOPE can improve over bad outputs. Moreover, judging from the presented results, bad early performance seems to undermine the overall accuracy (52%) a lot compared to full SCOPE accuracy (60%). More explanation and potential solutions would be necessary. However, given the additional experiments and clarification, I am inclined to raise my score.

**Key Questions For Authors:**

1. The method appears naive. What is the main challenge of training LLM-based models to predict CoT and utility metric? The proposed two stage training are also adopted from existing techniques.

2. The model performance depends on the anchor datasets. What is the size scale of this datasets B and token costs by pre-computing the results on 11 models? This matters because complicated queries impose additional costs, but smaller datasets undermine the retrieval performance.

3. Besides FLOPs, the actual inference time for a query is required, since the authors state that U_pred is obtained in real time, given that this directly affects the response time for user interaction.

4. The separation of prediction and decision is to adapt to different $\alpha$ preferences. What is the optimal $\alpha$ in practice? An approximate selection would suffice because usually users do not adjust such parameter in a dynamic manner for different queries. The experiments didn't seem to present a smooth change of $\alpha$ and the corresponding results.

Depending on the additional experiments and method clarification, it would be possible to adjust the rating.

**Limitations:**

See questions above

**Strengths And Weaknesses:**

1. The separation of metric prediction and model decision is novel, which allows fine-grained control by adjusting accuracy and cost preference.

2. Experiments are comprehensive, covering multiple models, baselines and datasets

---

> ### Author Rebuttal · Authors · 2026-03-31
>
> # Response to main challenges and SFT+RL novelty
> We thank the reviewer for the feedback. We agree that SFT and RL are now standard techniques. However, our main contribution is reformulating model routing as a reasoning problem, which required specific and targeted adaptations to these standard methods.
>
> **1. The Main Challenge: Learning a New Capability**
> The core challenge is teaching a model a new skill: inferring another model's future performance based on past evidence. Untrained models simply cannot do this. As shown in Table 2, the base Qwen4B model with 5 anchors performs poorly without training. This proves the task is non-trivial and justifies the need for our 60K training dataset.
>
> **2. Targeted Adaptations to SFT and RL**
> Because "predicting future behavior" has no natural reasoning steps, standard training pipelines fail. We introduced two key adaptations:
>
> - **Hindsight Distillation for SFT:** We conditioned a teacher model on realized outcomes to synthesize very short reasoning steps. This compresses the reasoning length from 2354.9 tokens down to just 238.7 tokens. This compressed initialization is critical: it speeds up the router and makes the later RL rollout process much more stable.
> - **Novel Reward Design:** In the RL (GRPO) stage, we replaced standard binary correctness with a gated composite reward. This explicitly targets both accuracy ($R_{corr}$) and token usage ($R_{token}$).
>
> **3. The Clear Impact of Each Stage**
> To explicitly show what SFT and RL contribute to our system, we provide the following ablation table:
>
>
> | Model Setting | Token MAE | Accuracy | Avg. Reasoning Length |
> | - | - | - | - |
> | Base Qwen3-4B | 1740      | 59.4%    | 2354.9 |
> | + SFT | 1754  | 59.0%    | 238.7 |
> | + SFT & RL    | 1555    | 77.0%    | 199.1 |
>
>
> This demonstrates our pipeline's logic: SFT handles the efficiency (compressing the trace), while RL handles the effectiveness (boosting accuracy and lowering error).
>
> # Response to anchor dataset size and cost
>
> We thank the reviewer for this important question. Balancing cost and retrieval quality is indeed key.
>
> **1. Anchor Set Size and Quality:** Our anchor set contains 250 representative queries (SCOPE-250). This compact size is rigorously validated to ensure quality:
>
> - **Figures 12 and 18:** Demonstrate broad and diverse semantic coverage.
> - **Figure 15:** Shows the category makeup perfectly mirrors the full 60K dataset.
> - **Figure 16:** Proves this small set accurately preserves the true token usage and accuracy behaviors of all models.
>
> **2. Low One-Time Pre-computing Cost:** Because the set is small, pre-computing model fingerprints is very cheap, as analyzed in Section 6.3:
>
> | Pre-computing Task     | Token Cost    |
> |-|-|
> | 1 Model (250 queries)  | ~1.22M tokens |
> | All 11 Models Combined | ~13.4M tokens |
>
> This one-time ~1.22M token overhead is much cheaper compared to the massive computational expense of retraining a routing model from scratch.
>
> # Response to actual inference time
>
> We thank the reviewer for this excellent point. We agree that FLOPs alone do not show the real waiting time for users, and reporting actual inference time is crucial.
>
> SCOPE's routing is fast for two reasons: predictions for each candidate model are independent and can run in parallel, and our highly compressed CoT makes the output very short.
>
> We measured the actual speed on a single 80GB A100 GPU:
>
> |Metric|Measurement (7-Model Pool)|
> |-|-|
> | **Wall-Clock Latency** | **0.54s** per query|
> | **Token Footprint**| **~1623 input + ~199 output** (**~1822 total**) per prediction |
> | **Throughput**| **~20K input + ~2.5K output** tokens/sec|
>
> For complex reasoning tasks where target models take tens of seconds to answer, a 0.54s delay is imperceptible. Furthermore, this tiny delay is frequently offset because SCOPE avoids slower, heavier models, often improving overall system latency instead of hurting it.
>
> # Response to $\alpha$ as a practical control knob
>
> We agree that end-users do not want to manually adjust parameters.
>
> **1. Automated, Not Manual Tuning:** Our separated design is built for automation. As detailed in Section 6.3, the "optimal" **α** is not a static value. Instead, the system automatically calculates the best **α** given the user's overall budget and targeted questions.
>
> **2. Evidence of Smooth Transitions:** We actually do demonstrate continuous, smooth changes across our experiments:
>
> - **Figure 6 (Left):** Displays a smooth Pareto frontier where SCOPE seamlessly navigates α.
> - **Figure 8:** Proves how the routing behavior smoothly transitions from cost-saving to accuracy-seeking as the user's budget relaxes.
> - **Figure 7 (Right):** The historical calibration weight **w** improves frontier coverage. Without calibration (**w**=0), routing decisions cluster at the low-cost end, leaving sparsity in the mid- to high-cost range. Adding **w=0.2** spreads selections more evenly across cost levels, yielding a smoother and more complete Pareto curve.

---

> > ### Author Rebuttal · Reviewer_1G5H · 2026-04-04
> >
> > 1. Given that the inference depends on heavy A100 GPU on 4B model, why not directly use this model for easy queries?
> > 2. In comparison, what happens if the target model is both effective and cheap like gemini pro series?
> > 3. Does this dataset include context reasoning? If we use cheaper models at early stage and output bad answers, would it affect the later performance? In other words, the overall performance in a series of conversation is not the summary of separate  models. The variance of the chose models may also incur bad outputs. How does SCOPE address this issue?

---

> > > ### Author Response · Authors · 2026-04-05
> > >
> > > # Q1
> > >
> > > We appreciate this excellent question, which touches on the core design philosophy of SCOPE.
> > >
> > > **First, identifying "easy queries" itself requires prediction.** Determining which queries are easy is precisely the predictor's role — without it, easy and hard queries are indistinguishable a priori. Once prediction is performed, SCOPE already achieves the reviewer's intended effect: the router naturally assigns easy queries to the cheapest candidate predicted to succeed.
> > >
> > > **Second, small models should serve as specialists rather than generalists.** An alternative might be letting the 4B model self-assess whether it can handle a query before prediction. However, prior work has shown that small LLMs exhibit systematic overconfidence across all difficulty levels [1], making self-evaluation unreliable. More fundamentally, a 4B model has limited capacity, and research demonstrates that SLMs should forgo generality to devote this capacity to a targeted task [2] — a small specialist scorer can even outperform large multi-task LLMs by focusing on a single objective [3].
> > >
> > > **Third, multi-model collaboration achieves more than any single model.** Rather than overburdening one small model with both prediction and generation, SCOPE lets each component do what it does best. This is the core design philosophy of SCOPE: routing itself is a form of multi-model collaboration, where a dedicated predictor and diverse candidates complement each other. As our experiments consistently demonstrate, this collaborative approach outperforms any single model acting alone.
> > >
> > > [1] Mind the Confidence Gap: Overconfidence, Calibration, and Distractor Effects in LLMs, arXiv:2502.11028.
> > > [2] Need a Small Specialized Language Model? Plan Early!, arXiv:2402.01093.
> > > [3] Cappy: Outperforming and Boosting Large Multi-Task LMs with a Small Scorer, arXiv:2311.06720.
> > >
> > > # Q2
> > >
> > > We conducted additional experiments by adding **Gemini 3.1 Pro** — a state-of-the-art 2026 model that significantly outperforms all 2025-era candidates in our pool. As a standalone model, it achieves **65.60% accuracy at $22.59**. We integrated it into SCOPE's candidate pool and further performed an analysis. Since Gemini 3.1 Pro is not actually cheap at its original pricing, we deliberately simulated the reviewer's hypothetical scenario — a model that is both effective and cheap — by artificially adjusting Gemini's price while using its actual token counts to estimate SCOPE's routing behavior:
> > >
> > > | Scenario | α=0 Acc | α=0 Cost | α=0.6 Acc | α=0.6 Cost | α=1 Acc | α=1 Cost |
> > > |-|-|-|-|-|-|-|
> > > | 5 models (Gemini@original *$2*/*$12*) | 37.60% | $0.62 | 49.60% | $1.47 | 62.00% | $3.81 |
> > > | +Gemini@DeepSeek (*$0.25*/*$0.38*) | 40.80% | $0.64 | 56.00% | $1.18 | 62.00% | $1.96 |
> > > | +Gemini@Near-free (*$0.001*/*$0.001*) | 65.60% | $0.00 | 68.40% | $0.47 | 62.00% | $1.69 |
> > >
> > > These results yield these key findings:
> > >
> > > **(1) SCOPE is effective even with a dominant model.** At original pricing, SCOPE achieves 62.00% accuracy at $3.81 (α=1) — an **83% cost reduction** compared to Gemini's standalone price. Moreover, SCOPE adapts dynamically to pricing changes: as Gemini's price decreases, the router automatically allocates more queries to it. This demonstrates that SCOPE dynamically exploits the best cost-performance trade-off available in the pool without any manual reconfiguration.
> > >
> > > **(2) Multi-model collaboration provides gains beyond any single model.** At near-free pricing with α=0.6, SCOPE achieves **68.40% — surpassing Gemini's standalone 65.60%**. This reflects a fundamental reality of current LLM ecosystem: no LLM is always the best. This is precisely why model routing matters, and why SCOPE's multi-model collaboration is effective — it captures complementary strengths across models that no single model, however dominant, can replicate alone.
> > >
> > > # Q3
> > >
> > > We thank the reviewer for raising this important concern. To directly address it, we conducted multi-turn experiments on CL-bench (10 multi-turn questions, 110 rubrics).
> > >
> > > We designed two strategies: (a) SCOPE: all turns freely routed by SCOPE; (b) forced_cheap+SCOPE: first 2 turns forced to gemma-3-27b (cheaper than all candidate models), remaining turns routed by SCOPE.
> > >
> > > | Strategy | Pass Rate | Total Cost | Cost vs. Claude |
> > > |-|-|-|-|
> > > | claude-sonnet-4.5 | 59.1% | $2.086 | — |
> > > | SCOPE | 60.0% | $0.176 | -91.6% |
> > > | deepseek-v3.2 | 50.0% | $0.119 | -94.3% |
> > > | forced_cheap+SCOPE | 52.7% | $0.027 | -98.7% |
> > >
> > > These findings suggest that (a) cheap early-turn context does not systematically degrade later performance; (b) SCOPE adaptively compensates by escalating to stronger models when needed; and (c) per-turn routing captures multi-model collaboration even in multi-turn settings. We note that the current results are achieved with an anchor set constructed from single-turn data. Reconstructing the anchor set with multi-turn conversational data to better capture turn-level dependencies is a promising direction for future work.

---

### Official Review · Reviewer_7K6y · 2026-03-13

**Soundness:** 2
**Presentation:** 3
**Significance:** 2
**Originality:** 2
**Overall Recommendation:** 4
**Confidence:** 4

**Summary:**

This paper addresses the limitations of static LLM routing systems by proposing SCOPE, a framework that shifts from fixed model selection to dynamic pre-hoc performance estimation. Instead of memorizing model IDs, SCOPE utilizes "behavioral fingerprints" derived from a 250-query anchor set to estimate the performance of unseen models without requiring retraining. The system employs a reasoning language model trained via SFT with hindsight distillation and GRPO to predict a candidate's binary correctness and token consumption before execution. Final routing decisions are made by maximizing a calibrated utility function that balances these predicted metrics based on a user-defined budget preference coefficient.

**Compliance With Llm Reviewing Policy:**

Affirmed.

**Final Justification:**

I have reviewed the follow-up responses and raised my scores accordingly. Please ensure that routing overhead is included in all efficiency evaluations in the next version.

**Key Questions For Authors:**

Please refer to the weakness

**Limitations:**

Please refer to the weakness

**Strengths And Weaknesses:**

Strengths

- The paper focuses on a very important question on LLM routing.
- The paper is well written and easy to follow.
- The proposed method replaces static model IDs with "behavioral fingerprints". By profiling a model's performance on a 250-query anchor set, it can dynamically route to new, unseen models without any retraining

Weaknesses

- While the application is interesting, the core training pipeline relies heavily on SFT and GRPO. Because the SFT + RL paradigm has become standard practice for training reasoning models, the purely algorithmic contributions of the paper feel somewhat incremental
- The baseline results are highly counterintuitive. The "Most Expensive" baseline severely underperforms SCOPE and is even beaten by a "Random" router on the OOD set. Furthermore, reporting a confusion matrix for a router would be beneficial to better understand the performance of each router, as the current router benchmark data is extremely skewed to certain model selection.
- While the paper claims cost efficiency, it is not explicitly clear if the reasoning costs for routing reported in Table 1 definitively account for the overhead of running the SCOPE router across the entire candidate pool before making a decision. The compared router baselines are considered as a very efficient router, where SCOPE routers may need an expensive reasoning process to conduct the routing process.
- Because SCOPE must run a pre-hoc evaluation for every candidate model in the pool, compute scales linearly with the number of models. More critically, the paper ignores wall-clock latency. Running multiple pre-hoc LLM evaluations introduces a non-trivial Time-To-First-Token delay that could render the system unusable for latency-sensitive applications.
- The utility function and decision calibration rely on seemingly arbitrary, hardcoded hyperparameters. There is no theoretical justification or empirical sensitivity analysis provided to prove these heuristics generalize beyond the paper's specific test setup.
- The proposed finite candidate search for the budget-controlled alpha requires solving piecewise affine intersections, mentioned in Appendix D. This feels over-selling and over-engineered for practical deployment, where a simple thresholding or binary search mechanism might achieve the same practical results with less mathematical overhead. Please correct me if I misunderstand the occasion.

---

> ### Author Rebuttal · Authors · 2026-03-31
>
> # Response to SFT + RL paradigm
> Thanks. Our core contribution is formulating model routing as a reasoning task, not proposing a new algorithm. Applying SFT+RL here required specific adaptations: we used hindsight distillation to create missing reasoning traces, and **compressed CoT lengths from 2354.9 to 238.7 tokens for efficiency** (mentioned in Sec. 6.3). For details, please see our **"Response to main challenges and SFT+RL novelty" to Reviewer 1G5H**.
>
> # Response to counterintuitive baselines & confusion matrix
> Thanks for your feedbacks.
>
> **Counterintuitive Baselines:** The "Random > Most Expensive" result on the OOD set happens because the candidate models are SOTA, and the questions are from very hard benchmarks. These frontier models have similar overall accuracy, but each excels at different types of questions. The most expensive model (Claude-Sonnet-4.5) does not necessarily outperform cheaper models like DeepSeek or GPT-5-mini on these tasks, making it less cost-effective.
>
> **Confusion Matrix:** We agree this is a great addition. Below, we define the "Ground Truth" (y-axis) as the **Cheapest Correct Model** (the oracle) versus the **SCOPE Selected Model** (x-axis) for α=1.0:
>
> |Oracle \ SCOPE Pred |Claude-4.5|DeepSeek-v3.2|GPT-5-mini|Grok-4.1-Fast|Total Oracle Selected|
> |-|-|-|-|-|-|
> |Claude-4.5|**7**|3|0|1|11|
> |DeepSeek-v3.2|38|**27**|36|32|133|
> |GPT-5-mini|13|5|**11**|5|34|
> |Grok-4.1-Fast|24|8|23|**17**|72|
> |Total SCOPE Selected|82|43|70|55|250|
>
> *(Note: The diagonal represents matching the absolutely cheapest correct model. Off-diagonal selections may still be correct, but use a more expensive model.)*
>
> This table clearly shows that our data is not skewed toward a single model. Instead, each model is chosen for the specific questions it is best at. Due to space constraints, we would be happy to provide more confusion matrix data in the follow-up discussion if the reviewer would find them helpful.
>
> # Response to reasoning costs
> Thanks. You are correct that the costs in Table 1 do not include the router's own reasoning overhead. However, this routing cost is practically negligible for two reasons:
>
> - **Local Deployment:** SCOPE uses a lightweight 4B open-source model. It can easily run locally, resulting in zero API costs.
> - **Highly Compressed Reasoning:** Even if using a commercial API, the cost is tiny. Our training forces the model to write very short reasoning steps (averaging only 199.1 output tokens). For the entire 250-query Test Set, the router processes ~2.84M input tokens and ~0.35M output tokens. At current Qwen3-4B API prices (*$0.004* per 1M input, *$0.017* per 1M output), the total routing overhead for all 250 queries is **$0.018**.
> # Response to wall-clock latency
> We thank the reviewer. The actual inference overhead of SCOPE is small: on a single 80GB A100 GPU, it adds only
> ∼
> ∼**0.54s** TTFT for a 7-model pool, enabled by parallel candidate evaluation and highly compressed CoT outputs. For details, please see our **"Response to actual inference time" to Reviewer 1G5H**.
>
> # Response to hyperparameter justification
> We thank the reviewer for this important point. SCOPE uses three main interpretable parameters: $\alpha$, which specifies the user's preference between accuracy and cost; $w$, which controls the strength of the historical calibration signal; and $\gamma_{\text{dyn}}$, which makes the normalized cost term comparable to accuracy in the function.
>
> For $w$, the paper already includes an ablation in **Fig. 7 (right)**, showing that moderate calibration yields the smoothest and most robust Pareto frontier.
>
> For $\gamma_{\text{dyn}}$, we further provide a sensitivity analysis:
>
> | $\gamma_{\text{dyn}}$ | Norm. AUC | $\Delta$ AUC |
> |:---:|:---:|:---:|
> | 2.0 | 0.6884 | -0.0039 |
> | **3.0** | **0.6923** | **ours** |
> | 4.0 | 0.6852 | -0.0071 |
> | 5.0 | 0.6784 | -0.0139 |
>
> The total normalized AUC variation is **below 0.02**, with $\gamma_{\text{dyn}}=3.0$ performing best. Overall, this suggests that SCOPE uses a small number of interpretable and robust parameters, rather than brittle heuristics tuned only for our specific setup. Due to space constraints, we would be happy to provide more analysis data in the follow-up discussion if the reviewer would find them helpful.
>
> # Response to $\alpha$ search
> We thank the reviewer for pointing this out.
>
> **Theoretical Proof, Not Operational Algorithm**: Appendix D is a theoretical justification. It proves that finding a budget-controlled $\alpha$ is mathematically well-posed and has a finite set of solutions, ensuring our method is not just an ad-hoc heuristic.
>
> **Simple Grid Search in Practice**: In our actual deployment, we do exactly as you suggest. We do not solve complex affine intersections online. Instead, we use a practical and simple grid search over $\alpha$ to find the best one for a given budget. In short, Appendix D provides the mathematical foundation, while the simple grid search is what we actually run.

---

> > ### Author Rebuttal · Reviewer_7K6y · 2026-04-03
> >
> > Thank you for your detailed responses to my questions. While I appreciate the clarification, I am still not fully convinced that the routing overhead is negligible in practice. The argument is based solely on token-level pricing, which does not capture system-level costs, such as latency and compute overhead, incurred by running a 4B model. In particular, it remains unclear how the routing cost scales with the size of the candidate pool, especially when reasoning across many candidates is required. Moreover, the comparison with efficient router baselines is not entirely fair, as those methods typically avoid generative reasoning and thus have fundamentally lower overhead. Without concrete latency or throughput measurements, it is difficult to assess whether the claimed efficiency gains hold in realistic deployment settings.
> >
> > Although the authors argue that the per-model prediction length is short after GRPO training, they still report an average of 238.7 prediction tokens per model per query, which is not obviously negligible when compared against efficient classifier-based routers that do not perform generative reasoning over every candidate. Moreover, Table 1 reports routing performance and cost, but the rebuttal itself admits that these numbers do not include the router’s own reasoning overhead, so the comparison with efficient routing baselines is not fully apples-to-apples. Thus, I lean to remain my origainl score.

---

> > > ### Author Response · Authors · 2026-04-04
> > >
> > > We sincerely thank the reviewer for the continued discussion. We respectfully note that several concerns raised in the final response appear to have been directly addressed in our rebuttal. As permitted and recommended by ICML guidelines under the 5,000-character limit, we consolidated shared concerns into unified responses with cross-references to other reviewers. We recognize this format may have made some answers less immediately visible, and we apologize for the inconvenience. To ensure full clarity, we reproduce and expand the key data below.
> > >
> > > # Concern 1: "Without concrete latency or throughput measurements..."
> > >
> > > We provided these measurements in our rebuttal under “Response to wall-clock latency” and "Response to actual inference time" to Reviewer 1G5H. We reproduce the table here for convenience:
> > >
> > > | Metric | Measurement (7-Model Pool) |
> > > | :--- | :--- |
> > > | **Wall-Clock Latency** | **0.54s** per query |
> > > | **Token Footprint** | **~1623 input** + **~199 output** (**~1822 total**) per prediction |
> > > | **Throughput** | **~20K input + ~2.5K** output tokens/sec |
> > >
> > > All measurements were taken on a single 80GB A100 GPU.
> > >
> > > Regarding scalability: SCOPE evaluates each candidate model independently, so reasoning traces for all *N* candidates are generated in a single batched forward pass. We verified the scaling behavior empirically:
> > >
> > > | Candidate Pool Size | Wall-Clock Latency |
> > > | :--- | :--- |
> > > | 3 | 0.31s ± 0.09s |
> > > | 5 | 0.38s ± 0.09s |
> > > | 7 | 0.54s ± 0.08s |
> > > | 9 | 0.89s ± 0.06s |
> > > | 11 | 1.48s ± 0.09s |
> > >
> > > For typical candidate pools (5-7 models), the latency remains under 0.6s. Even at 11 models, the 1.48s overhead is modest compared to the target model's own inference time, which often takes tens of seconds for complex reasoning tasks. Moreover, this overhead is frequently offset by SCOPE steering queries away from slower, heavier models.
> > >
> > > # Concern 2: Routing cost not included in Table 1
> > >
> > > We respectfully argue that adding reasoning overhead to Table 1 would conflate two distinct metrics. Table 1 reports *target model API cost*, measuring routing quality — how well a router reduces the cost of calling target models. This is the standard metric all baselines are evaluated on, and keeping router overhead separate is precisely what ensures an apples-to-apples comparison. The router's own inference overhead is a separate dimension and should be compared independently.
> > >
> > > That said, we fully agree with the reviewer that routing overhead deserves a transparent comparison. We use wall-clock latency as the comparison metric, as it captures the total end-to-end cost of each router — whether that cost comes from an embedding model forward pass (as in classifier-based routers), multi-turn LLM interaction (as the xRouter in Table 1), or batched single-turn generation (as in SCOPE). We benchmarked all routers on the same hardware (single A100 GPU):
> > >
> > > | Router | Latency per Query |
> > > | :--- | :--- |
> > > | MLP Router | 0.048s |
> > > | KNN Router | 0.053s |
> > > | xRouter | 18.6s |
> > > | SCOPE (Ours) | 0.54s |
> > >
> > > SCOPE is slower than classifier-based routers, which is expected given that they avoid generative reasoning. However, the gap is modest in absolute terms (0.54s vs ~0.05s), and both are minor relative to target model inference. Notably, xRouter, which also performs reasoning over candidates, incurs 34x higher latency than SCOPE, highlighting the efficiency of our compressed CoT and parallelized evaluation design.
> > >
> > > # Summary
> > >
> > > We have taken concrete steps to ensure SCOPE is a deployable system: compressed CoT reasoning (from 2354.9 to 199.1 tokens), parallelized candidate evaluation, and empirically verified low latency. Our results demonstrate strong routing performance with overhead that is practical for real-world deployment. We will incorporate all the above measurements and tables into the revised manuscript.

---

### Decision · Program_Chairs · 2026-04-30

**Decision:**

Accept (regular)

**Comment:**

This paper proposes SCOPE, a dynamic LLM routing framework that replaces static model selection with pre-hoc performance estimation using behavioral fingerprints derived from a compact anchor set. The framework enables generalization to unseen models without retraining, and offers controllable accuracy–cost trade-offs via a user-specified preference coefficient. The problem is timely and practically important, and the empirical results are strong across both in-distribution and out-of-distribution settings.

Key concerns raised — including routing latency overhead, the necessity of LLM-based reasoning over simpler predictors, the contribution of the RL stage, … — were addressed in the rebuttal with concrete experiments and technical discussions. The authors are encouraged to incorporate these additional results and comparisons into the final version.